**Experimental production of charcoal morphologies to discriminate fuel source and fire type: an example from Siberian taiga**

Angelica Feurdean[1]

5 [1] Department of Physical Geography, Goethe University, Altenhöferallee 1, 60438 Frankfurt am Main, Germany

*Correspondence to*: angelica.feurdean@gmail.com; Feurdean@em.uni-frankfurt.de

**Abstract**

The analysis of charcoal fragments in peat and lake sediments is the most widely used approach used to reconstruct past biomass burning. With a few exceptions, this method typically relies on the quantification of the total charcoal content of the sediment. To enhance charcoal analyses for the reconstruction of past fire regimes, and to make the method more relevant to studies of both plant evolution and fire management, the extraction of more information from charcoal particles is critical. Here, I used a muffle oven to burn seven fuel types comprising 17 species from boreal Siberia (near Teguldet village) which are also commonly found in the Northern Hemisphere, and built on published schemes to develop morphometric and finer diagnostic classifications of the experimentally charred particles. I then combined these results with those from fossil charcoal from a peat core taken from the same location to demonstrates the relevance of these experiments to the fossil charcoal records. Results show that graminoids, *Sphagnum*, and wood (trunk) lose the most mass at low burn temperatures (<300°C), whereas heathland shrub leaves, brown moss, and ferns at high burn temperatures. This suggests that species with low mass retention in high-temperature fires are likely to be underrepresented in the fossil charcoal record. The charcoal particle aspect ratio appeared to be the strongest indicator of the fuel type burnt. Graminoid charcoal particles are most elongate (6.7-11.5), with a threshold above 6 that may be indicative of wetland graminoids, leaves are the shortest and bulkiest (2.0-3.5) while twigs and wood are intermediate (2.0-5.2). Further, the use of fine diagnostic features was more successful in separating wood, graminoids, and leaves, but it was difficult to further differentiate these fuel types due to overlapping values. High aspect ratio particles, dominated by graminoid and *Sphagnum* morphologies, may be robust indicators of low temperature, surface fires, whereas abundant wood and leaf morphologies, and low aspect ratio particles are indicative of higher-temperature fires. However, the overlapping morphologies of leaves and wood from trees and shrubs make it hard to distinguish between high-intensity surface fires, combusting living shrubs and dead wood and leaves, and high-intensity crown fires which have burnt living trees. Despite these limitations, the combined use of charred-particle aspect ratios and fuel morphotypes can aid the more robust interpretation of changes in fuel source and fire type. Lastly, I highlight the further investigation needed to refine the histories of past wildfires.

## 1 Introduction

Disturbance by wildfires is among the most common disturbance types in boreal forests, triggering gap dynamics or stand-scale forest replacement depending on intensity (temperature of fire) and frequency (Goldammer, 2015). Ongoing and anticipating increase in the intensity and frequency of wildfire in boreal forests is raising concerns on its impact on the composition of these forests as well as climate (Jones et al., 2020). Although most of the global boreal forest area is in Siberia (Furayev, 1996), its vast extent and restricted access have limited datasets recording changes in wildfire activity, especially from a longer-term perspective (Marlon et al., 2016). Such long-term records of wildfire activity are vital to understanding how fire regimes vary with changes in climate and human-vegetation interaction, as well as the impacts of fires on boreal forests.

Wildfires reach temperatures up to 1800 °C, however, charcoal is an inorganic carbon compound resulting from the incomplete combustion of plant tissues, which typically occurs at temperatures of 280–500 °C (Rein, 2014). Charcoal particles vary in size and form, but can preserve characteristics such as edge aspects, surface features, cleavage, lustre, or anatomical details (tracheids with border pits, leaf veins, cuticles, etc.) that can be used to determine the origin of the fuel (Ward and Hardy,

1991; MacDonald et al., 1991; Scott 2010; Enache and Cumming, 2006; Jensen et al., 2007; Courtney-Mustaphi and Pisaric, 2014a; Hubau et al., 2012, 2013; Prince et al., 2018). Although macroscopic charcoal analysis, typically counting charcoal pieces or charcoal area per unit sediment volume from a small sediment volume (1-2 cm$^3$), is widely applied, the full potential of this method has not been explored. This analytical limitation restricts the reconstruction of fuel sources, a crucial factor in determining fire type, i.e., the burning of surface fuels in low or high intensity fires or distinguishing between surface and

crown fires. At a minimum, this requires a broader distinction of the morphology of charcoal to indicate the nature of plant material burnt (Courtney-Mustaphi and Pisaric, 2014 a,b; Feurdean et al., 2017; Hawthorne et al., 2018). Furthermore, the determination of fire type is not only critical to palaeofire reconstructions, but it is an accessible tool for ecosystem managers and modellers, and for assessing and mitigating the risks of fires that might impact settlements and infrastructures (Moritz et al., 2014).

Ongoing efforts have advanced the utility of charcoal morphological analyses for fuel type identification and fire regime reconstruction. Umbanhowar and McGrath (1998), Crawford and Belcher (2014), and Pereboom et al. (2020) conducted morphometric measurements of the length, aspect ratio (length/width), and surface area of charcoal particles by burning in the laboratory known plant materials originating from American prairie, tropical and arctic environments and concluded that longer fragments correspond to graminoids, whereas shorter fragments originate from wood, shrubs, and leaves. Nichols

(2000) and Crawford and Belcher (2014) additionally found that charcoal morphometrics is generally preserved during transportation by water. Other studies have focused on the effects of burning conditions, i.e., open-flame ignition, muffle furnace experiments (Umbanhowar and McGrath, 1998; Orvis et al., 2005), and combustion calorimetry (Hudspith et al., 2018) on charcoal production. Blecher et al. (2005, 2015) investigated in the laboratory whether fire can be ignited by thermal radiation and be the reason for major extinction events in deep geological times, with results giving little support for this

hypothesis. Jensen et al. (2005) and Courtney-Mustaphi and Pisaric (2014a) examined subtler diagnostic features (morphology, surface features, lustre) of laboratory-produced charcoal morphotypes of a small number of North American grasses and leaves of coniferous and deciduous trees. Furthermore, Enache and Cumming (2006, 2007) and Mustaphi and Pisaric (2014a) classified charcoal morphologies in Canadian lake sediments based on particle shape (morphology), aspect ratio, and surface features, and linked them to fuel types. Courtney Mustaphi and Pisaric (2014a) also discussed the potential for categorising of

charcoal morphologies to explore relationships to taphonomy processes and fuel types. Based on published morphotype categorisations, more recent studies have attributed fossil charred particles to certain fuel and fire types (Walsh et al., 2010, Daniau et al., 2013; Aleman et al., 2013; Courtney-Mustaphi and Pisaric, 2014a,b, 2018; Feurdean et al., 2017, 2019a,b, 2020; Unkelbach et al., 2018).

This paper presents the first results of laboratory-produced charcoal morphologies (muffle oven) spanning a range of fuel types originating from 17 boreal Siberian species. It aims to characterise the diversity of charcoal morphologies produced by boreal understory and forest vegetation to facilitate more robust interpretations of fuel sources in the study region. Specifically, it evaluates (i) whether morphological distinctions (morphometrics and finer anatomical features) exist between species and fuel types, (ii) the effect of burning temperature on the mass, morphometrics, and finer anatomical features of charred plant material, and (iii) discusses the advantages and limitations of laboratory-based burning studies for palaeofire reconstruction. Because this combination of factors has never been tested to such an extent in the laboratory, this study has the potential to significantly advance our understanding of the link between sedimentary charcoal morphologies and fire types which will substantially refine wildfire histories in boreal forests. As most of the species used in this study are common to Northern Hemisphere forests and peatlands, these results are directly applicable over a broad geographical scale.

## 2 Material and Methods

### 2.1 Laboratory analysis

Plant materials used for laboratory burning experiments were identified in the field, stored in plastic bags for transportation, and air-dried. Selected materials include a range of fuel types (graminoid, trunk wood, twigs of tree and shrub, and the leaves of coniferous and deciduous trees, shrubs, forbs, and ferns, moss, and fern stems with leaves) from the most common tree, shrub, herb, fern, and moss species around a forested bog near Teguldet village, Tomsk district, Russia. This light boreal taiga forest is primarily composed of *Pinus* and *Betula.* Additionally, needles and twigs of *Picea abies* were collected from Taunus, near Frankfurt am Main, Germany (see Table 1 for a full list of species and fuel types). All plant material collected was from living plants, except trunk wood, which was taken from a dead tree.

To determine the mass, morphometrics, and finer diagnostic features of residual charred plant material, as well the effect of increasing temperatures on all these characteristics, dried remains of individual plant species were placed in ceramic crucibles, weighed, covered with a lid to limit oxygen availability, and heated for 2 h in a muffle oven (preheated for 1 h) then roasted at 250, 300, 350, 400, or 450 °C (File S1). No burning experiments were conducted at higher temperatures because all plant material turned to ash i.e., a solid residue mostly composed of minerals that crumbled apart into soot and flyash (Rein et al., 2014). It also tested the effect of mixing plant material in known ratios on charred mass and morphometrics at 300 °C, an intermediate temperature. For this, the plant material was combined in the following volume to approximate the predominant fuel mixtures for low-intensity surface fires (75% graminoid and moss: 25% shrub; 50% graminoid: 50% moss and fern), intermediate- to high-intensity surface fires (25% graminoid and moss: 75% shrub; 50% graminoid and moss: 50% shrub), and high-intensity crown fires (50% graminoid, shrub, and moss: 50% wood and leaf). The experimental temperatures were chosen based on the range of temperatures reported in the literature (250–500 °C; Umbanhower and McGrath, 1998; Orvis et al., 2005; Jensen et al., 2007; Pereboom et al., 2020). It should be noted that dry roasting in a muffle oven approximates some aspects of the heating conditions of vegetation in a natural fire. However, a crucial variable that is not explored is the time at a burning temperature and oxygen conditions (Belcher et al., 2015; Hudspith et al., 2017). Additionally, with roasting in an

oven, the influence of flame dynamics and turbulent airflow is reduced, and therefore plant tissue is more rapidly reduced to ash than in natural fires.

After cooling each sample, the remaining charred mass was weighed and calculated the ratio of charred to pre-combustion mass. Charred samples were then split into two subsets. The first was left intact and stored as reference material. The second was gently disaggregated with a mortar and pestle to mimic the natural breakage that charcoal particles would incur overtime in sediment (Umbanhowar and McGrath, 1998; Crawford and Belcher, 2014; Belcher et al., 2015), then washed through a 125-μm sieve to remove smaller fragments. This second sample split was subjected to both morphometric measurements and

characterization of finer diagnostic features. Morphometric measurements of individual charred particles were obtained from photographs taken at 4× magnification with a digital camera (Kern DXM 1200F). On average, more than 100 charcoal particles larger than 150 μm were automatically detected in most samples, except for those burnt at higher temperatures, where particles were more prone to breaking up. The major ($L$) and minor ($W$) axes, along with surface area ($A$) of each particle were measured following the algorithm presented in Appendix A1, and calculated aspect ratio as $L/W$. Finer diagnostic features such as shape,

surface features (reticulates, tracheids with border pits, leaf veins, the arrangement of epidermal cells, cuticles with stomata, etc), and cleavage were characterized at $4 \times$ by inspection of microphotographs or the charred particles themselves under a microscope or stereomicroscope.

To demonstrate the applicably of these experiments to the fossil records, seven samples with higher charcoal content were selected from a sediment core from Ulukh-Chayakh mire near Teguldet village (Feurdean et al., in prep). The preparation of

the samples followed Feurdean et al. (2020) and includes bleaching overnight and washing in a 150-μm sieve. The results were compared to pollen and plant macrofossils data from the same core.

### 2.2. Numerical analysis

The medians and standard deviations of charcoal morphometrics ($L$, $L/W$, $A$) were aggregated for each species, fuel type, and

burn temperature, and are displayed as box plots. The two-tailed Mann-Whitney test was used to test whether the medians of the charcoal morphometrics of various fuel types were equal (File S2). This test does not assume a normal distribution, only equally-shaped distributions in both groups.

### 3. Results

### 3.1 The influence of temperature on charred mass production

Only a few fuel types (needles, shrub leaves) were greenish or brownish in colour at 250 °C. Plant materials of all species were black with a typical charcoal appearance after burning at 300 and 350 °C. A few fuel types (graminoid, *Sphagnum,* and some twigs) turned to ash at 400 °C, whereas all remaining plant tissue became ash at 450 °C (File S1). Most of the charred materials remained intact and retained all their morphological characteristics. However, those burnt at higher temperatures tended to

easily break during sample manipulation.

The average percentage of charred mass retained at 300 °C (an intermediate temperature) was as follows in decreasing order: brown moss and fern (50%) > wood (shrub twig 46%) > leaf (shrub 44%) > leaf (forb 42%) >leaf (needles 41%) > wood (tree twig 40%) > graminoid (29%) > *Sphagnum* (22%) > wood (trunk 11%; Table 2; Fig. 1). This trend in the mass loss was largely the same at all temperatures, with charcoal mass showing a marked decline from 38-84 % at 250°C to 0.2-23% at 400°C across

all fuel types. The charred mass of mixed-fuel samples at 300 °C was lowest for samples with high contents of graminoid and *Sphagnum* (33-35%) and highest for samples with greater proportions of the shrub (38%).

### 3.2 Fuel-dependent variations in length, aspect ratio, and surface area

Graminoid charcoals burnt at 300 °C ($L/W$ = 11.5, Fig. 2b, f, Table 2) were consistently more elongate than those of twigs

(shrub, 5.2; tree, 3.8), moss and fern stems (4.6), and leaves (2.7). Among leaves, charred needles were more elongate (3.1) than those of forb (3.3), heathland shrub (2.3), and broadleaf tree leaves (2.1). The Mann-Whitney test confirmed that (i) the median aspect ratio of graminoids was significantly different from those of all other fuel types ($p < 0.001$), (ii) those of woods (all types) were different from those of leaves ($p < 0.001$) and moss (except at 350 °C) and (iii) those of leaves were different from those of moss ($p < 0.001$; Table S2). In contrast to the median aspect ratio, the lengths (major axis, $L$) of charred particles

from different fuel types were less clearly differentiated (Figs. 3a, c, Files S2, S3). The surface area ($A$) varied greatly between individual taxa and fuel types, however, fragments of shrub leaves tended to be larger than all other fuel types (Figs. 3b, d, Table 2, Files S2, S4). The morphometrics of mixed-fuel samples showed that charcoals of samples with abundant graminoids and moss were more elongate (higher $L/W$) than those with higher proportions of shrubs, wood, and/or leaves (Fig. 2h). Similarly, the longest charcoal particles (higher $L$) were from samples with greater proportions of graminoids and moss (Fig.

3e), whereas the charcoals with the largest surface areas were from samples with more abundant shrubs and leaves (Fig. 3f). The aspect ratios and lengths of individual taxa and fuel types changed slightly with temperature, but the general trends were similar across all temperatures (Figs. 2, 3a, c, File S3). In contrast, relative surface areas varied more with temperature changes (Figs. 3b, d, File S4).

**3.3 Finer diagnostic features of the charcoal morphologies of various fuel types**

**3.3.1 Graminoid charcoal**

Graminoid (*Carex, Calamagrostis, E. vaginatum*) charred particles were consistently flat, rectangular, and elongated (Figs. 4a, A2a). They mostly broke parallel to the long axis when pressured, resulting in highly elongated pieces with straight margins. They can also appear as featureless long, thin filaments. Charcoals produced at higher temperatures (350 °C) often had more

irregular, zig-zag, or denticulate margins. Their surface features more commonly preserved rectangular epidermal cells or contained oval voids, reticulated or mesh patterns, and/or isolated veins.

**3.3.2 Wood charcoal (trunk, tree and shrub twigs)**

Wood charcoal pieces from trunk (*P. sylvestris*) were blocky and quadrilateral with corner angles of 90° (Figs. 4b, A2b). Wood charcoal from tree (*P. sylvestris, P. sibirica, Picea abies, B. pendula)* and shrub (Ericaceae) twigs showed both quadrilateral and polygonal shapes. For both trunks and twigs, edges were smooth, serrated, or denticulate, and surface textures were smooth, foliated, or striated (Fig. 4b). Trunk charcoals of *P. sylvestris* showed rows of brown, open pits in the tracheid walls. Under the microscope, trunk charcoal fragments were shinier and darker than twig charcoals. Large charcoal pieces often broke parallel to the long axis, producing many tiny, elongated pieces (trunks) or pieces of various forms (twigs).

### 3.3.3 Leaf charcoal (needles, deciduous tree and shrub, forb, and fern)

Charred needle fragments were elongated and rectangular (corner angles of 90°; Figs. 4c, A2a). Their edges were smooth but became serrated and denticulate when broken. Surface features included visible venation and ridges. Charcoals from the leaves of deciduous trees (*Betula*), heathland shrubs (*Oxyccocum, Ledum, Camadaphne, Vaccinium*), herbaceous plants (*Rubus),* and ferns (Polypodiaceae) were polygonal. Only those of *Cnidium* leaves were elongated, reflecting their needle shape. Edges were mostly undulate, but sometimes smooth or denticulate. Surface textures were generally smooth (featureless), but sometimes included visible venation and ridges. When broken, they showed voids, reticulated mesh patterns, and curly fibres. Birch leaves produced visible charred veins with three branches diverging from a node. When pressured and broken, small leaf pieces had fracture lines radiating out at a variety of angles.

### 3.3.4 Moss and fern stems

*Sphagnum* produced two types of charcoal morphologies. One type originates from stems were elongate with ramifications (scars) where leaves branched from the stem, and the other originates from leaves preserved the anatomical features of the unburned leaves, i.e., a mesh-like appearance (Figs. 4a, A2a). *Polytrichum* produced several charcoal morphologies (quadrilateral, polygonal, or curved with angular edges) with generally featureless surfaces, although some showed mesh patterns. This charcoal type often splits along the main axis. *Equisetum* leaves and stems were generally quadrilateral with straight, undulate, or denticulate margins, and oval voids and reticulated mesh patterns on their surfaces.

### 3.4 Morphometrics and finer diagnostic features of fossil charcoal

The average aspect ratio of the seven sediment samples was 35 cm =3.2, 84 cm = 3.0, 85 cm = 3.0, 172 cm = 4.0, 248 cm = 11.1, 268 cm = 4.3, and 302 cm =2.8 (Table 3). Samples with higher aspect ratios contained abundant morphologies of graminoids, *Equisetum* and moss (Table 3). The average surface area was 35 cm =248.287, 84 cm = 122.599, 85 cm = 219.413, 172 cm = 49.673, 248 cm = 16.565, 268 cm = 53.3354, and 302 cm =53.698, where samples with greater surface area have higher number of leaves.

### 4 Discussion

Results from the current laboratory burning experiments allow the characterisation of charcoal morphology assemblages i.e., morphometrical aspects, fine diagnostic features, and charcoal production for 17 plant species belonging to seven fuel types from boreal Siberia. This dataset substantially broadens the geographical coverage of fuel types researched, demonstrates the applicability of charcoal morphology assemblages to fossil records, and improves the interpretation of fire types based on charcoal morphologies.

## 4.1 The influence of combustion temperature on charcoal production: implications for charcoal-based fire reconstructions

Knowledge of the charred mass is critical for determining biases in charcoal production to biomass quantity and fire temperature (Walsh and Li, 1964). Results from these burning experiments clearly show that the effect of temperature on charcoal production is fuel dependent. Graminoid, *Sphagnum*, and trunk wood produce the lowest amounts of charcoal per unit biomass and lost their mass more rapidly with increasing burning temperature i.e., from 40-63% at 250°C to 0.2-3% at 400°C (Fig. 1; Table 2). Contrastingly, leaves of heathland shrubs, forbs, and ferns (Polypodiaceae), as well as fern stems (*Equisetum*), produced the most charcoal per unit biomass and retained the greatest mass at higher temperatures i.e., from 50-84% at 250°C to 4-24% at 400°C (Fig. 1). The charred mass of mixed-fuel samples also changed according to the dominant fuel type (at 300 °C). Peerboom et al. (2020) burned plant tissue from similar taxa occurring in the Alaskan tundra in a muffle oven (at 500 °C) and likewise found that graminoids have a lower charred-mass (25-27%) retention than shrubs (up to 33%). However, they neither tested the leaves and wood of shrubs separately, nor the effect of various temperatures on charred mass. In partial agreement with findings from this study, burning experiments of American forest steppe plants in a muffle oven (at 350°C) and under open flame conditions showed that the mass retention of grass and deciduous leaf charcoal decrease more rapidly with temperature compared to wood charcoal (Umbanhower and McGrath, 1998). More recent calorimetric combustion of various fuel types from plants mostly originating from tropical taxa found the charred mass of wood, needles, and *Equisetum* to be greater than other leaf types, connected to higher bulk densities and fuel load (Hudspith et al., 2017).

Although burn conditions in the current oven experiments do not fully replicate those of natural wildfires, the findings present some practical implications for charcoal-based fire reconstructions. First, fuel types with low charred-mass retention at hotter burn temperatures might be underrepresented in the sedimentary charcoal record. Specifically, Cyperaceae (sedges) are the most common graminoids in fens and meso- and eutrophic bogs worldwide, and *Eriophorum* (sedge) and *Sphagnum* (moss) are common in oligotrophic bogs. These fuel types are likely to be the first to turn to ash, even in relatively low-intensity fires (<300°C), and thus may leave little or no trace of charcoal in sediments. Second, woody biomass of *Vaccinium, Camadaphne, Oxycoccus,* and *Ledum*, typical heathland shrubs of oligotrophic bogs, and *Polytrichum commune* (brown moss) common to all habitat types, preserve almost half of their biomass up to 300 °C. However, their masses decline strongly (11-15%) at higher temperatures (>350°C), indicating that they are likely to be preserved as charcoal only in low- to intermediate-intensity fires. Third, leaves of shrub, forb, and fern, and stems with leaves of *Equisetum* are more likely to persist as charcoal (27-38%) after high-temperature fires and thus may contribute disproportionately to sedimentary charcoal. Mineral constituents can slow

pyrolysis (thermal decomposition of plant material producing volatile products and a solid charred residue) and this is probably why *Equisetum* stems with high silica content, preserve more charcoal. Fuel with higher lignin content (wood) should also produce more charcoal than fuels higher in cellulose and hemicellulose i.e., leaves (Yang et al., 2007). In contrast to this expectation, leaves in the present laboratory experiments, retained a higher charred mass than the wood with increasing burn temperature. This calls for extending research on the quantitative relationship between temperature and charcoal mass retention

to fuels with various structures, chemistry, and bulk density.

**4.2 Fuel-dependent variability in charcoal morphometrics: implications for the reconstruction of fuel-type and transportation by air and water**

Some of the charred fragments produced in this study show consistent morphometrics among species within the same genus

and family, suggesting their utility for fuel-type identification. Graminoid charcoal particles are at least two times more elongated (6.7-11.5) than all other charcoal types and differ the most from leaf charcoals across all temperatures (Fig. 2; Table 2; File S5). Highly elongate and narrow graminoid charcoals are posited as resulting from the occurrence of conspicuous veins parallel to the long axis (Umbanhowar and McGrath, 1998; Crawford and Belcher, 2014). Charred fragments of leaves (2.0-2.7 broadleaves; 3.1-3.5 needles) are also markedly more circular than those of other fuel types. However, there is some degree

of overlap between the aspect ratios of twigs (2.5-5.2), and moss and fern stems (3.5-4.75; Table 2). In agreement with Crawford and Belcher (2014) and Umbanhowar and McGrath (1998), we found that the smaller the particles, the lower the aspect ratio (more circular particles). Although larger charcoal fragments may be more suitable to categorise fuel type, it is difficult to define a threshold aspect ratio concerning the size of the particles to be used for such measurements. Charcoal fragments from mixed-fuel samples also preserve the aspect ratio of the dominant fuel type; particles with highest aspect ratios

(3.5) were found in samples with greater proportions of graminoids and moss. In contrast, there is less confidence in using the length and mean surface area to distinguish between fuel types, except for the slight tendency that charred shrub leaf particles are larger than those of all other fuel types (Figs. 3, File S4; Table 2). The larger shrub leaf fragments may be explained by the arrangement of leaf venation, with fragments breaking along the three branching veins that diverge from nodes (Umbanhowar and McGrath, 1998; Jenssen et al., 2007).

The aspect ratio of graminoids and shrub and forb leaves observed in this study are most similar to those from the Alaskan arctic, where graminoids (*Eriophorum vaginatum* and *Carex bigelowii*) show aspect ratios ranging from 5.46 to 8.09 (mean 6.77), and shrubs (*Ledum palustre, Salix pulchra, Betula nana, Rubus chamaemorus, Vaccinium vitis-idaea*) from 2.09 to 2.50 (mean 2.42; Pereboom et al., 2020; Table 4). Considerably shorter graminoid particles were obtain from American steppe forests i.e., 3.62, however, aspect ratios were closer to those in this study for leaves 1.91 and wood 2.13 (Umbanhowar and

McGrath, 1998). Under laboratory conditions Crawford and Belcher (2014) produced charcoal with an aspect ratio of 3.7 for graminoids, 2.23 for leaves, 1.97 for wood, and 2.8 for *Pinus sylvestris* needles (Table 4). Fossil charcoal assemblages from tropical African forests and grasslands were used to separate graminoids (aspect ratio <2.0) from shrubs (>2.0; Aleman et al., 2013), whereas Daniau et al. (2013) employed the rise in the aspect ratio as an indication of the increased proportion of burning

of the grass fuel. Mustaphi and Pisaric (2014a) also observed that burning monocotyledons from boreal Canada in the laboratory generally produced more elongated charcoal morphologies than other fuels. In term of surface area, Umbanhowar and McGrath (1998), show a surface area of 65.630 $\mu m^2$ (56.737 herein) for graminoids, 50.150 $\mu m^2$ (103.138 herein) for wood and 64.946 $\mu m^2$ (versus 114.952 herein) for leaves, comparable to Pereboom et al. (2020) who found little differentiation between average surface area for shrub (88.246 $\mu m^2$), and graminoid (87.474 $\mu m^2$) species. Results from current measurements of the effect of temperature on charred-particle morphometrics show that the aspect ratio is generally preserved for all fuel types over the temperatures explored. However, the length and surface area of fuel types changed less consistently with increasing temperature. Umbanhowar and McGrath, (1998) found that burn temperature did not significantly change the aspect ratios of graminoid and leaf charcoals, but that it marginally reduced those of wood.

The combined results from this study with those from published literature suggest that, despite some variability in morphometrics of charcoal assemblages from similar fuel types, a decreasing aspect ratio from graminoids to wood and leaves appears to be upheld in most studies. However, only graminoids consistently display a substantially high aspect ratio to enable their separation from other fuel types. Based on the mean aspect ratios of the three graminoid species used here (6.7-11.5), the threshold aspect ratio indicative of graminoids could be set above 6. It should be noted, however, that this value averages a wide range of individual measurements. Nevertheless, this threshold value may be valid for wetland graminoids (mean 6.77 for artic Alaska) but may be too high for graminoids from temperate grasslands (3.8-4.66; Table 4). Although there is also a good consistency in the aspect ratio of laboratory-produced wood (2.1-4.5) and leaf (2.0-3.5) charcoal particles across studies, these values strongly overlap (Tables 2, 4). It is therefore not possible to specify a threshold value at which charcoal particles are indicative of wood or leaves. The use of charcoal morphologies in fuel type identification, therefore, requires the use of fine anatomical features (see section 4.3) or validation from other sources such as anthracological analysis as employed in archaeobotanical studies (Hubau et al., 2012; 2013; 2015; de Melo Júnior, 2017).

The shape of charcoal particles affects the behaviour of charcoal during transportation by air (Clark, 1998; Clark and Hussey, 1996) and water (Nicols et al., 2000). Models, assuming a uniform spherical particle shape, and empirical data of transportation by fume indicate that the amount of charcoal particle is greatest near the fire source (Clark et al., 1998; Clark and Royall, 1995; Tinner et al., 2006; Higuera et al., 2007; Peters and Higuera, 2007). However, recent models accounting for different shapes, sizes, and densities of charcoal show that non-spherical particles have lower settling velocities than spherical particles and produce a spatially more extensive and heterogeneous particle-size distribution pattern, i.e, dispersal distances for spherical and aspherical particles greater than 150 µm could be up to 20 km apart (Vacula and Richter 2018). Similarly, Clark and Hussey (1996) derived a velocity index for sedimentary charcoal particles and found that non-spherical particles have lower setting velocities and higher residence time into the atmosphere than the elongated particles. Based on these studies, it appears that non-spherical charcoal particles (elongated) such as those of graminoid, moss, and fern are likely to have a more heterogeneous distribution and be deposited farther away from the origin of a fire than the rounder, polygonal leaf particles. Erosion during hydrological transportation can also change the shape of buried (sedimentary) charcoal and can be an important consideration when interpreting charcoal morphometrics (Patterson et al., 1998; Nichols et al., 2000; Scott et al., 2000).

Laboratory experiments simulating fluvial transportation found that the surface area of leaf charcoal decreases and circularity increases with transportation, whereas changes in the shape of woody particles is less evident with transportation, and grassy charcoal preserves a high aspect ratio during transportation (Crawford and Belcher, 2014). However, Nichols et al (2000), found a slight rounding of sharp-angled edges of wood and a greater propensity for breakage of charcoal produced at higher temperatures. These findings give further support that the typical appearance of graminoids as elongated particles and of leaves as circular is preserved during transportation. Nevertheless, other studies using sedimentary charcoal records suggest that erosion during transport accounts for the rounding (degree angles are eroded) of robust charcoal types such as wood, whereas fragile pieces of leaves and grass may break (Vanniere et al., 2003; Mustaphi and Pisaric, 2014b; Mustaphi, et al., 2015). The differential transportation by air and fragility of sedimentary charcoal morphotypes calls for investigations for the influence of particle shape on charcoal transportation and strategies targeting coring locations for generating robust quantitative data for palaeofire interpretations.

**4.3 Finer diagnostic features of the charcoal morphologies for fuel type identification**

Results from fine diagnostic features on charcoal particles show that these can be more confidently used to attribute charcoal particles to certain fuel types. Apart from the extremely elongated shape that differentiates graminoid charred particles from all other fuel types, graminoids are further distinguished under both microscope and stereomicroscope by their flat appearance and breakage into thin filaments (Figs. 4a, A2a). Rectangular epidermal cells, reticulate meshes, oval voids of former epidermal stomata are also good diagnostic features of graminoids (Grosse-Brauckman, 1974). The graminoid charcoals produced in this study are most similar to types C4, C6, D1, D2, and D3 described by Mustaphi and Pisaric (2014a) and Enache et al. (2006). Comparative studies on graminoid charcoal originating from Poaceae (grass) versus Cyperaceae (sedge) family will further improve the identification of fuel types given the ecological differences of the two groups i.e., sedges growing on wetlands, and grass often on dry habitats.

A distinct feature of woody charcoal is that they are layered with foliated or striated textures and break into many tiny particles when pressured (Figs. 4b, A2b). This is due to the abundance of fibres and xylem, which leads to charcoals splitting at various angles (Vaughan and Nichols, 1995). Additionally, conifer wood charcoal presents distinct rows of open pits in the tracheid walls (Schweingruber, 1978). Attempts to distinguish between charred trunk and twig particles were less successful, although charred trunk particles are blockier. Foliated charred wood fragments also share appearance with moss and fern stems. These woody charcoals are most similar to types A1, B1, B2, and B3 (Mustaphi and Pisaric, 2014a; Enache et al., 2006).

Typical features of charred deciduous leaves are their polygonal shapes with surfaces characterised by void spaces or undulated surfaces (Figs. 4c, A2a). Netted venation is also sometimes visible, mostly with three branches diverging from a node. In contrast, conifer needles are elongated, often show ramification, and can have a wood-like appearance. The deciduous leaf charcoals found here are most similar to morphologies A2, A3, A4, A5, and A46, and conifer needle charcoals to C1, C2, and C3 (Mustaphi and Pisaric, 2014a; Enache et al., 2006).

Charred *Sphagnum* leaves preserve the meshed pattern of fresh plant material (Grosse-Brauckman 1972). Often, stems contain ramification, likely scars of former leaves (Figs. 4a, A2a). Both *Sphagnum* and *Polytrichum* charcoals present curvy fragments not seen in other fuel types. However, stems of *Sphagnum* and *Polytrichum* can be easily be mistaken for shrub twigs. Burnt *Equisetum* can resemble graminoid charcoal. Charred moss is similar to morphologies C4 and C7 (Mustaphi and Pisaric, 2014a; Enache et al., 2006).

### 4.4 The morphometrics and morphologies of fossil charcoal particles

Charcoal fragments from Holocene samples ranging from 6700 to 180 cal yr BP at the Ulukh-Chayakh mire preserved the aspect ratio of the dominant morphologies i.e., particles with the highest aspect ratios (4-11) were found in samples with a greater proportion of graminoids, *Equisetum,* and moss (Table 3). Likewise, a greater surface area was found in samples with a higher number of leaves. Comparative results from fossil charcoal morphologies and morphometrics to those from pollen and plant macrofossils from the same depths show a partial agreement. For example, the pollen record indicates that percentages of tree were >90% and shrub up to 3% during the entire period, whereas the abundance of woody charcoal morphologies increased infrequently. This suggests that although there was a continuous source of woody fuel to burn, high intensity fire, producing wood charcoal occurred only occasionally. There is, however, a better agreement between samples with greater aspect ratios and morphologies of understory vegetation i.e., graminoids, *Equisetum,* and moss, and the proportion of these plants in the pollen and plant macrofossil records (Table 3). These findings are in line with the Siberian wildfire behaviour of predominantly low intensity, surface fires fuelled by graminoids, forbs, ferns, and mosses, or intermediate intensity surface fires (shrubs) and only infrequently as high intensity crown fires (Anderson, 1982). Another practical application of this finding is that the morphometrical and morphological characterisation of fossil charcoal is more representative of fuel type (what plant types were burning), whereas the pollen data and plant macrofossils reflect plant types growing regionally and locally.

### 4.5 Applications

The physical and chemical characteristics of fuel are key factors influencing ignition and fire propagation. Major chemical components of fuels are cellulose, hemicellulose, and lignin, and minor ones include terpenes, resins, and minerals (Plana and Pastor, 2014). Fuels rich in cellulose and hemicellulose (i.e., leaves) pyrolyse at a lower and narrow temperature range (200 and 400 °C), whereas those rich in lignin (i.e., wood) pyrolyse at a higher and over a wider range of temperatures (160-900°C; Yang et al., 2007). Fuel types rich in terpene and resins (conifer wood, needles, Ericaceae) burn faster and hotter, whereas those rich in mineral components (graminoids) burn at lower temperatures (200 °C) and less efficiently (Plana and Pastor, 2014). Results from the current burning experiments and fossil charcoal samples suggests that the combined use of morphometric and morphological features and charred mass can help distinguish some of the predominant fuel source. Knowledge of the fuel source may in turn provide clues on fire type, i.e., the combination of fire intensity (temperature) and severity (effect on vegetation). Fossils samples dominated by graminoid morphotypes show a high aspect ratio (4 -11), in line

with the elongated shape of graminoid charcoal found in burning experiments (>6.7-11.5). As graminoid charcoal typically preserves at lower temperatures, it likely suggests a graminoid fuel source, and therefore a lower-intensity fire (Fig. 5). Fossil samples with abundant leaves and wood morphologies showed considerably lower aspect ratios (3-3.2), in agreement with values from laboratory-derived morphologies of leaves (2.0-3.5) and wood (2.0-5.2). Thus, shorter, and bulkier charcoal particles likely indicate the increased prevalence of leaves and wood as a fuel source (Figs. 2, 5). Because the morphometrical and morphological characteristics of leaves, and wood from trees and shrubs overlap, it is hard to distinguish between high-intensity surface fires, combusting living shrubs and dead wood and leaves, and high-intensity crown fires which have burnt living trees. Nevertheless, the fact that the past fires may have been of higher intensity at times of leaves and wood charcoal dominance than during the graminoids dominance is additionally suggested by the occurrence of *Equisetum* and *Polytrichum,* taxa found to remain as charcoal after burning at high temperature.

In summary, the good consistency of results from this study with those from the literature on various vegetation types (boreal, temperate, and tropical woodlands, and grasslands) suggests the potential of charcoal morphometrics and morphologies in palaeoecology. For example, the expansion of open habitats during deep geological times or with human impact, the recession of latitudinal and elevational treelines, or the occurrence of surface fires is likely to be reflected in an increase in aspect ratio and graminoid morphologies relative to total biomass burning. Conversely, the closing up of the forests, shrub encroachment, or the predominance of crown fires may show itself in a decreased aspect ratio of particles and increased bulky morphologies derived from leaves and wood. Results could also provide forest managers with the range of fire types that key boreal species experienced in the past, useful when aiming to make choices for prescribed burning to remove fuel and prevent large fires or select species that will be fit to cope with future fire regimes. Answering all questions, however, will need further investigations to relate the proportion of charcoal morphotypes to the quantity of biomass and extend the morphometric and morphological characterisation to key species of interest.

## 5. Conclusions and recommendations

This study presents the first results of the morphometric aspects and finer diagnostic features of charred particles produced in the laboratory from seven fuel types comprising 17 plant species from boreal Siberia and demonstrates the applicably of these experiments to the fossil charcoal records. The use of a higher number of fuel types from species with broad geographical coverage combined with an exploration of various combustion temperatures improves the link between charcoal morphologies, fuel types, and fire characteristics. Results show a distinct effect of temperature on fuel types, suggesting that species with low mass retention (graminoid, *Sphagnum*, and trunk wood) during high fire temperature are likely to be underrepresented in the fossil charcoal record. Among morphometrics, the aspect ratio was found to be the strongest indicator of fuel type. Graminoid charcoal particles are more elongate (6.7-11.5) than all other fuel types with a threshold above 6 that may be indicative of wetland graminoids, leaves are the shortest and bulkiest (2.0-3.5), and twigs and wood are intermediate (2.0-5.2). Further, the use of fine diagnostic features was more successful in separating wood, graminoids, and leaves, but due to overlapping values, it is hard to make further distinctions within these fuel types.

Despite these limitations, the combined use of particle aspect ratio and charred morphotypes allows more robust interpretations
of changes in fuel source and fire type based on charcoal records. Future efforts to determine fuel sources based on analyses
of small charcoal fragments will require: i) a more detailed examination of plant anatomy; ii) relate the proportion of particular
charcoal morphotype to the quantity of biomass; iii) quantify the relationship between the chemical composition of fuels,
combustion temperature, and charcoal production; iv) determine the influence of particle shape on differential transportation
and fragility; and v) use of image-recognition software to collect data on other charcoal features such as roundedness (the
440 degree angles), reflectance and other features may improve estimations of fire temperature and erosion during transportation.

**Data sets:** A limited amount of burnt plant material can be made available upon request.

**Author contribution**: AF designed the burning experiments and carried them out. AF performed the morphometrical and
morphological analyses; AF performed numerical analysis and data presentation; AF wrote the manuscript; AF acquired the
445 financial support for the project leading to this publication.

 **Competing interests**: The author declare that they have no conflict of interest.

Special issue statement**:** The role of fire in the Earth system: understanding interactions with the land, atmosphere, and
society (ESD/ACP/BG/GMD/NHESS inter-journal SI)

**Acknowledgements** I would like to thank Markus Rosensthil for the help to develop the code for automatic detection of
450 charred particles and drawing the pictograms in Fig 5; Dagmar Fritzsch for initial brainstorming on the burning experiments;
Doris Schneider for help with burning plant material in muffle oven; and Sergey Kirpotin for help with identification of plant
species in field.

**Financial support:** This work was supported from the Deutche Forschungsgemeinschaft grant number FE_1096/6.

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

**Figure 1.** The percent of charred mass retained after burning known plant species from Siberia in a muffle oven at 250, 300, 350, and 400 °C. Abbreviations: L, leaf; N, needles; t, twig; w, wood. The median mass retained for similar fuel types (identified by the same colour) are reported as black diamonds.

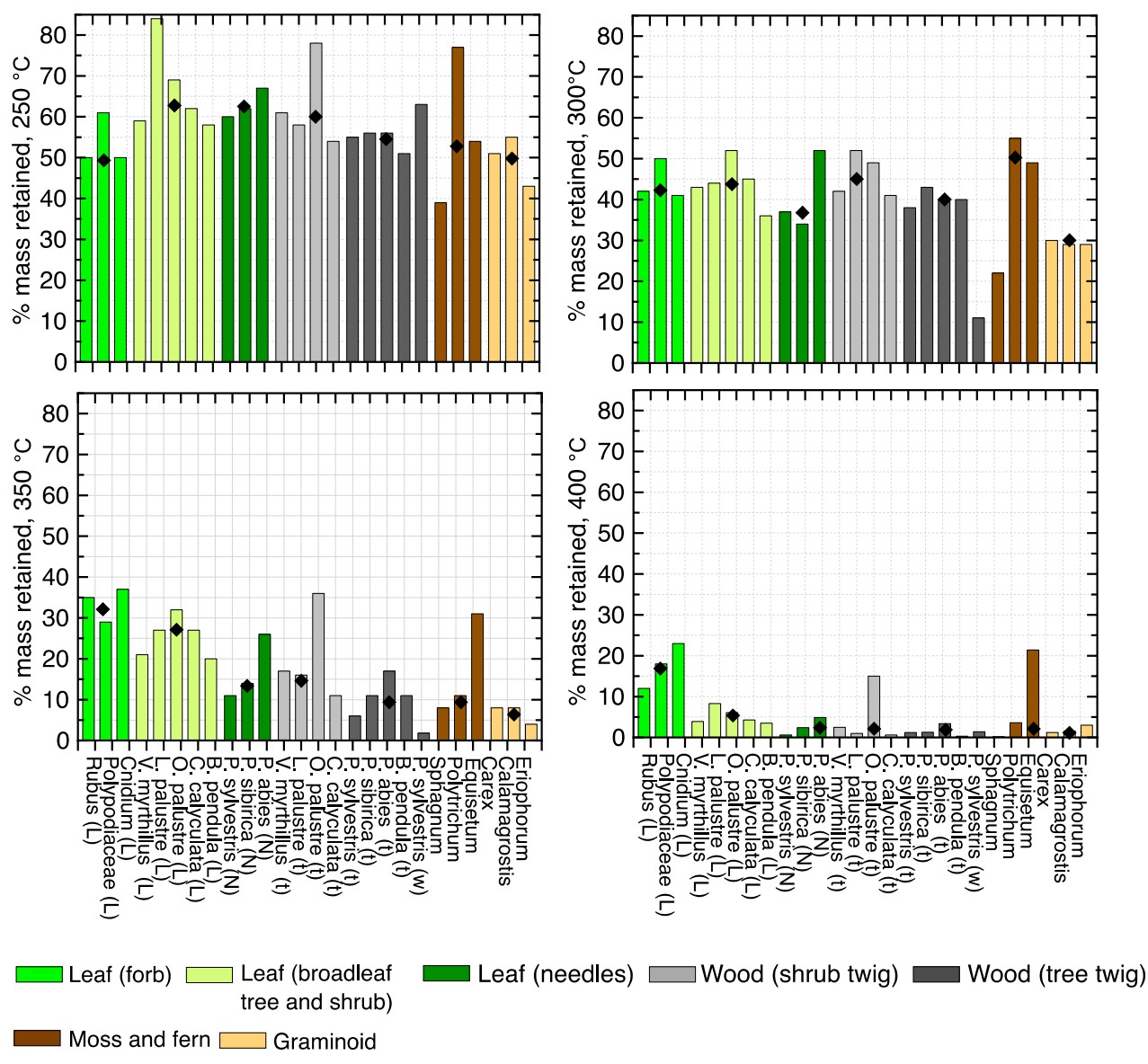


**Figure 2.** The median aspect ratios of charred particles from (a–d) the individual measurements burned at 250, 300, 350, and 400 °C, respectively, and (e–g) fuel types at burning temperatures of 250, 300, and 350 °C, respectively, as well as from (h) mixed-fuel samples burned at 300 °C. The fuel mixtures are arranged in order of increasing proportions of graminoids. Box plots represent the distribution of data as follows: the horizontal line in each box denotes the median, the upper quartile is the median value of the upper half of the data points, the lower quartile is the median value of the lower half of the data points, whiskers represent the minimum and the maximum values. Abbreviations of plant material burned are given in Figure 1. The individual taxa belonging (a-c) to a fuel-type group (d-g) are indicated by the same colour.

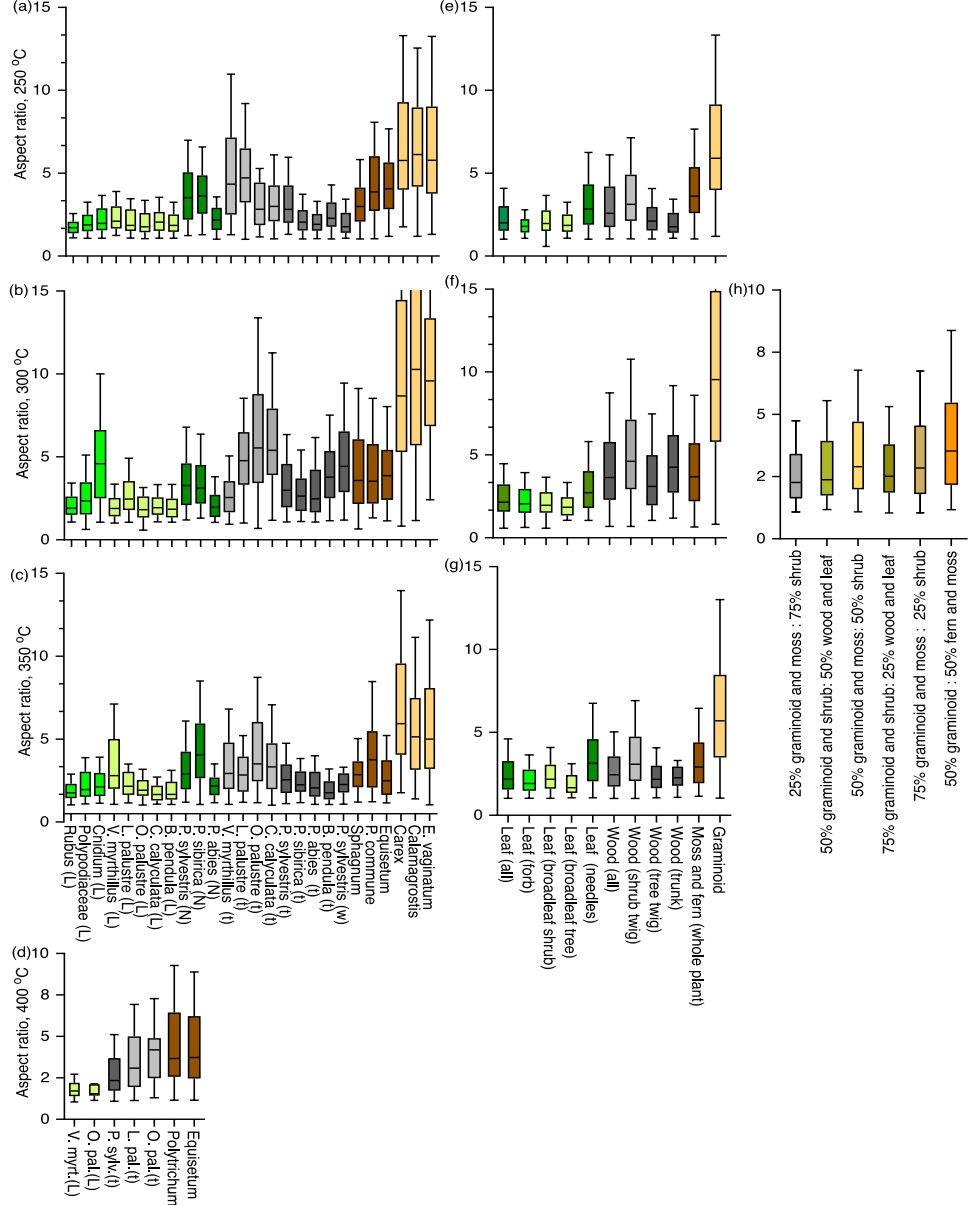

**Figure 3.** The median lengths (µm) and surface areas (µm²) of charred particles from (a, b) individual taxa, (c, d) fuel types, and (e, f) fuel mixtures at 300 °C. Abbreviations as in Figure 1. See Figure 2 for description of box plots and colour coding.


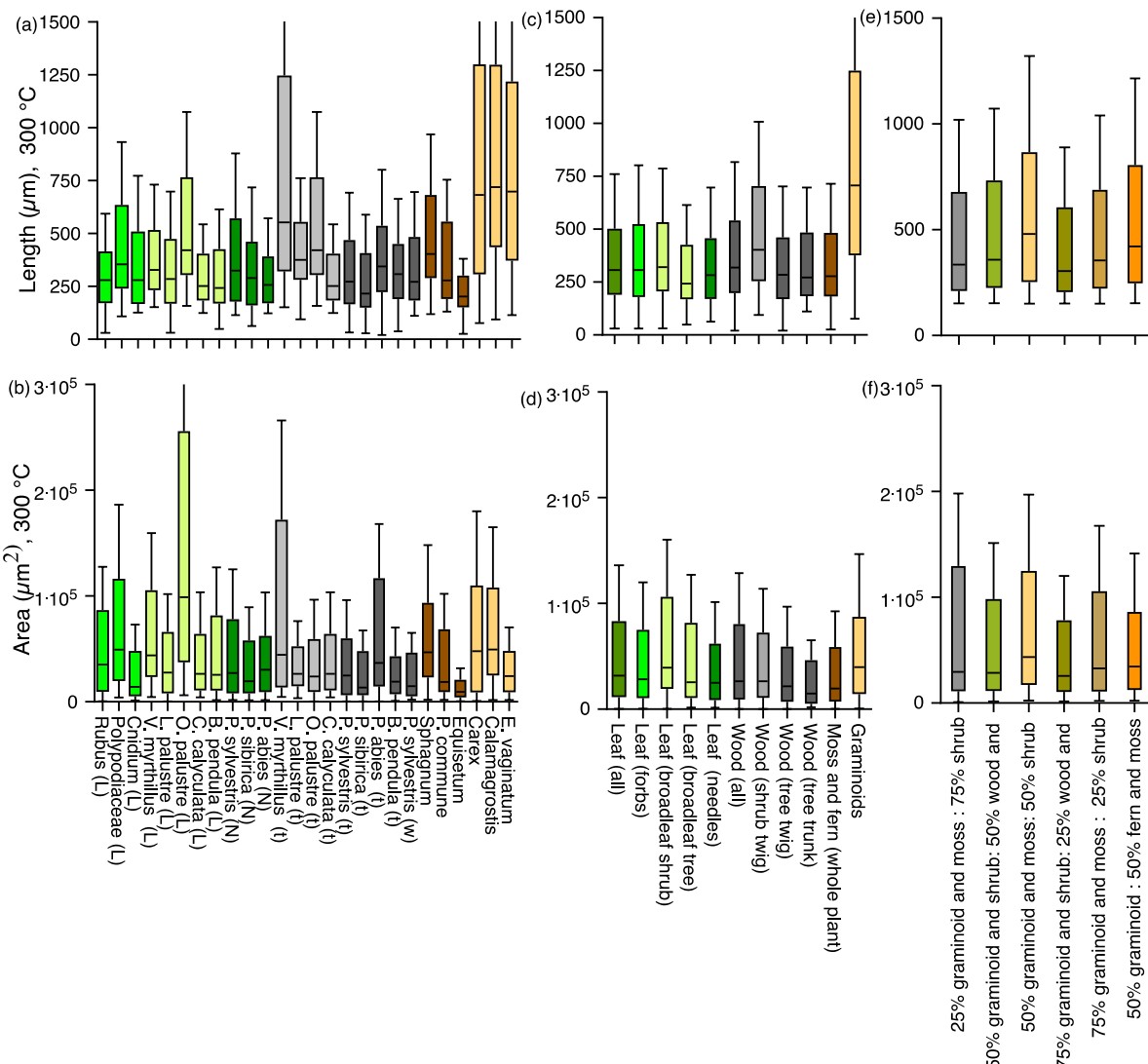


**Figure 4.** Photomicrographs of characteristic charcoal morphotypes under stereomicroscope (4 ×). (a) Graminoids (1–10), ferns (11–13), and moss (14–18). (b) Wood from tree twigs (1–9), shrub twigs (10–15), and trunks (16–20). (c) Conifer needles (1–6), deciduous tree leaves (7–9), shrub leaves (10–15), fern leaves (16–18), and moss leaves (19–20).

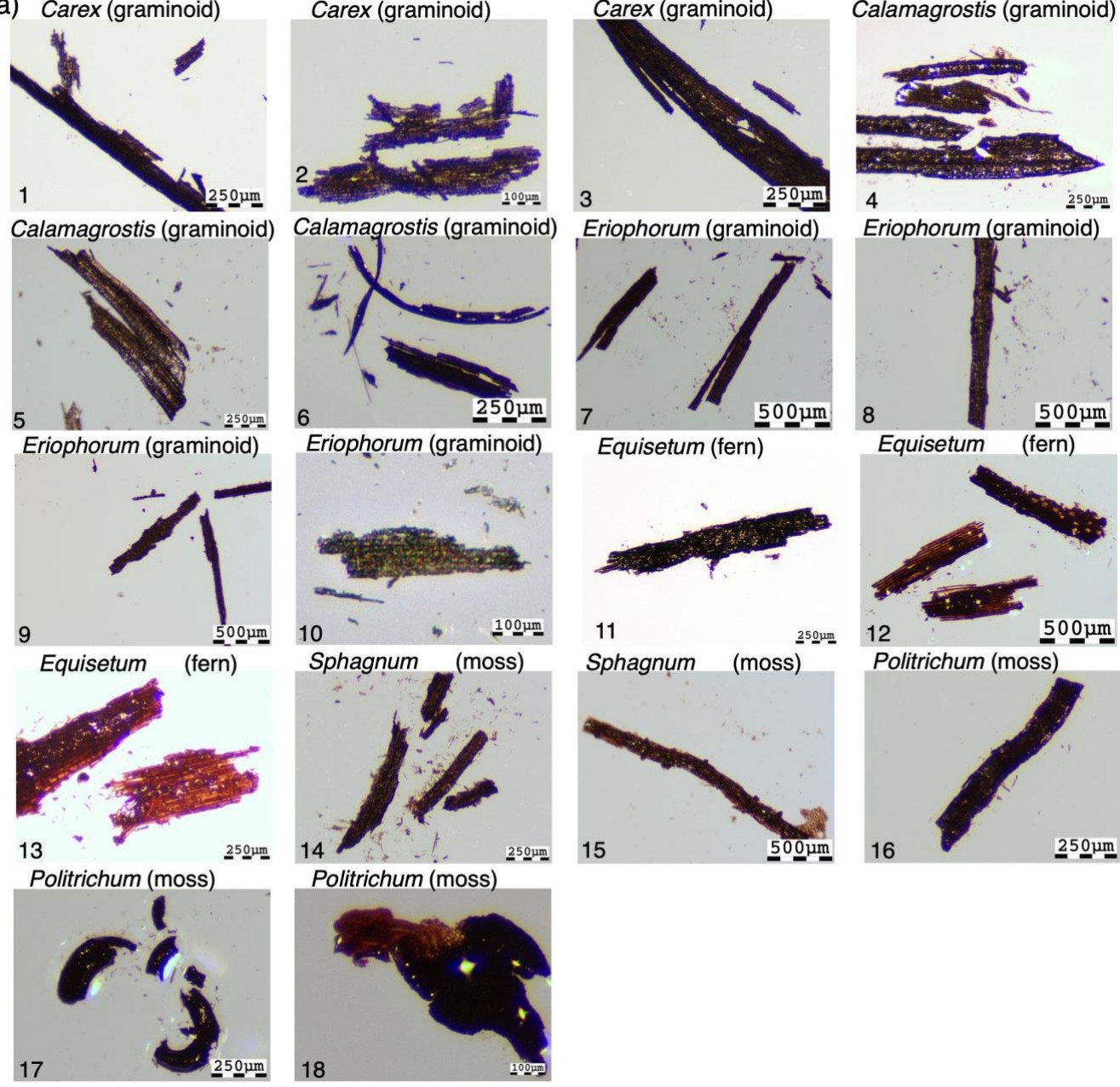


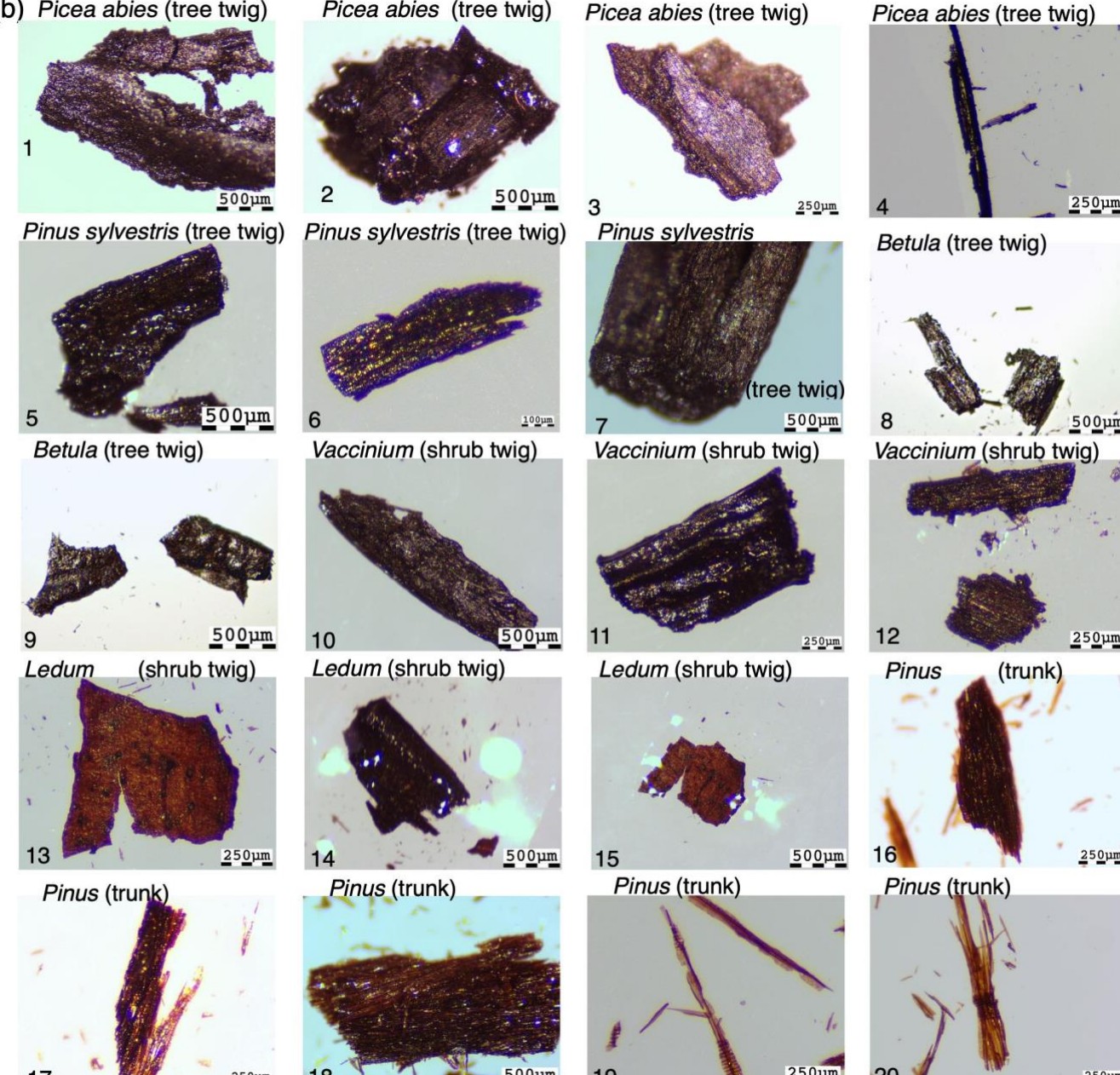

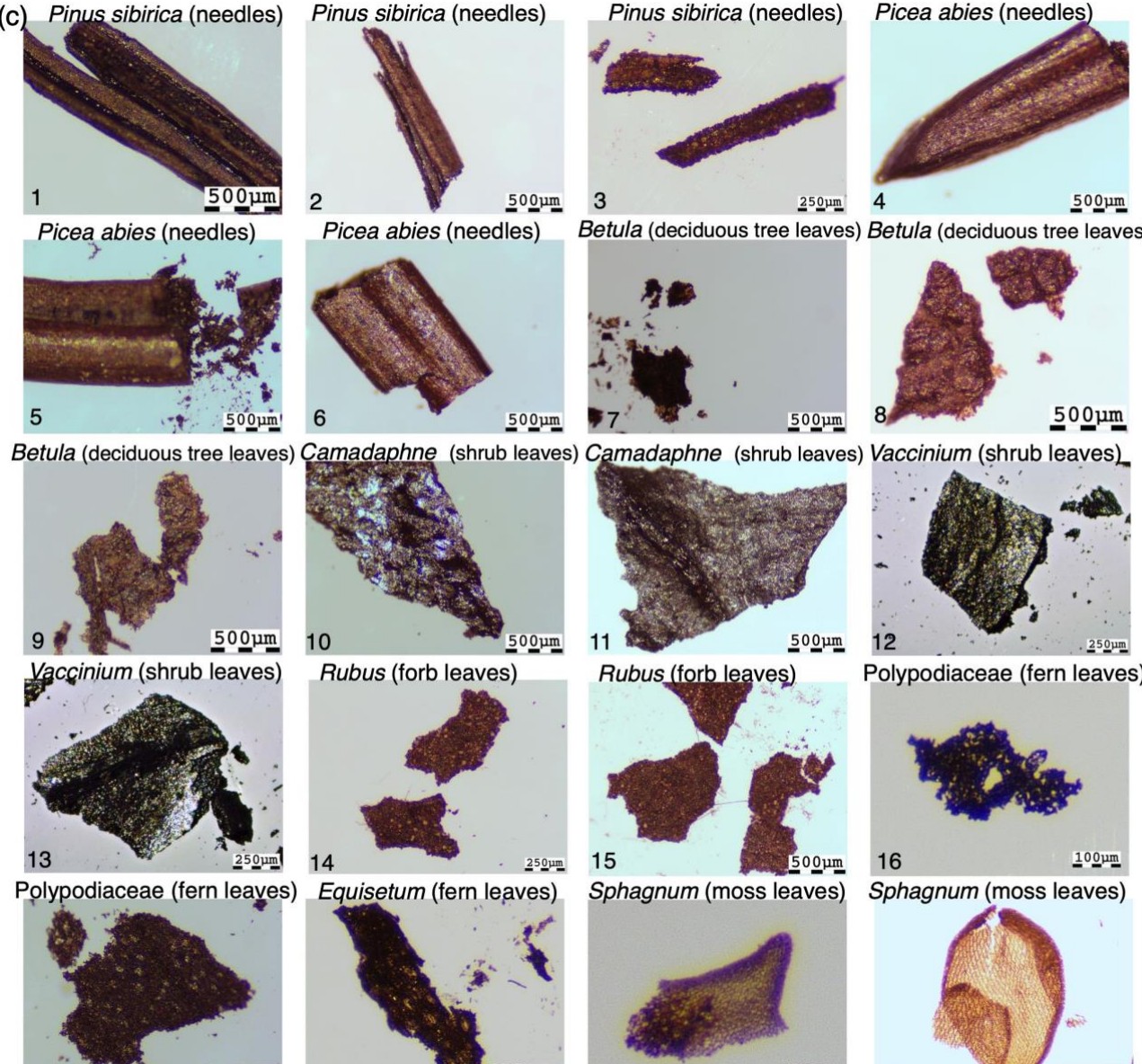


**Figure 5.** Schematic representation of fire types and the potential link with fuel types burnt and predominant charcoal morphometrics (aspect ratio) and morphologies as well as charcoal production.

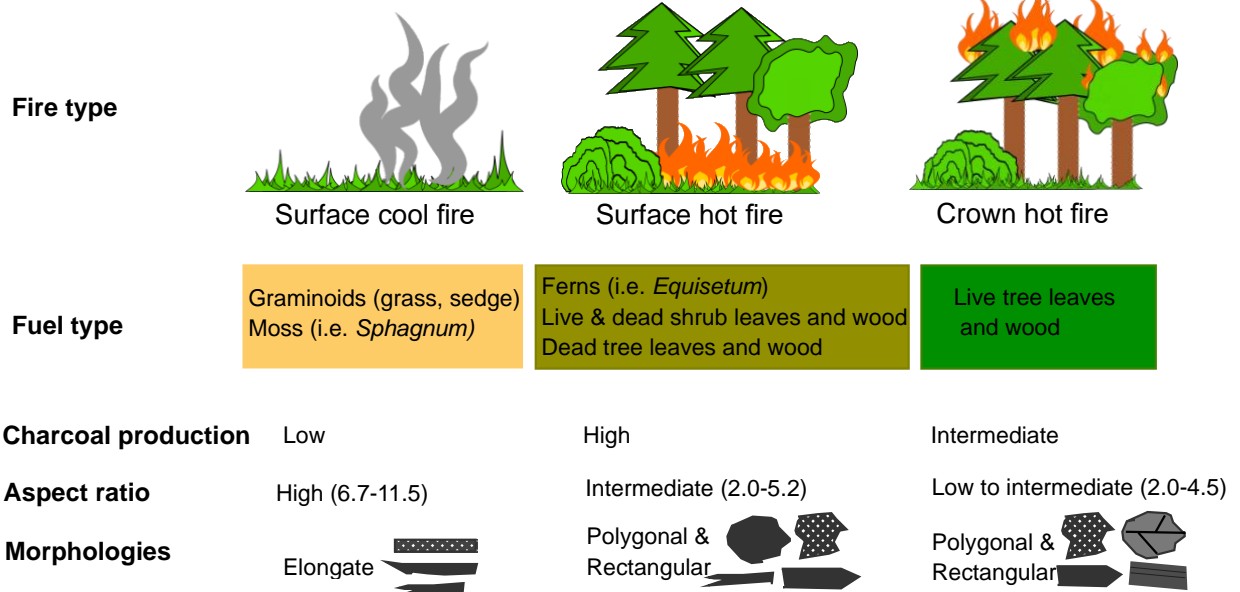

**Tables**

**Table 1.** List of plant materials burned. All plants are from Siberia, except for *Picea abies*, originating from Taunus, Germany. All plant material was dried before the combustion in the muffle oven. Leaves of deciduous trees and shrubs include veins and petioles. The term twig was used only for woody species i.e., deciduous and coniferous trees and shrubs and no distinction was made between soft young wood.

| Plant type | Scientific name | Family | Common name | Plant burned |
|---|---|---|---|---|
| **Trees** | | | | |
| Conifer tree | *Pinus sylvestris* | Pinaceae | Scots pine | Needles |
| Conifer tree | *Pinus sylvestris* | Pinaceae | Scots pine | Twigs |
| Conifer tree | *Pinus sylvestris* | Pinaceae | Scots pine | Dead wood |
| Conifer tree | *Pinus sibirica* | Pinaceae | Siberian pine | Needles |
| Conifer tree | *Pinus sibirica* | Pinaceae | Siberian pine | Twigs |
| Conifer tree | *Picea abies* | Pinaceae | Norway spruce | Needles |
| Conifer tree | *Picea abies* | Pinaceae | Norway spruce | Twigs |
| Deciduous tree | *Betula pendula* | Betulaceae | Silver birch | Leaves |
| Deciduous tree | *Betula pendula* | Betulaceae | Silver birch | Twigs |
| | | | | |
| **Shrubs** | | | | |
| Shrub | *Vaccinium myrtillus* | Ericaceae | Bilberry | Leaves |
| Shrub | *Vaccinium myrtillus* | Ericaceae | Bilberry | Twigs |
| Shrub | *Oxycoccus palustre* | Ericaceae | Bilberry | Leaves |
| Shrub | *Oxycoccus palustre* | Ericaceae | Bilberry | Twigs |
| Shrub | *Empetrum  nigrum* | Ericaceae | Bilberry | Leaves |
| Shrub | *Empetrum nigrum* | Ericaceae | Bilberry | Twigs |
| Shrub | *Ledum palustre* | Ericaceae | Bilberry | Leaves |
| Shrub | *Ledum palustre* | Ericaceae | Bilberry | Twigs |
| Shrub | *Chamaedaphne calyculata* | Ericaceae | Bilberry | Leaves |
| Shrub | *Chamaedaphne calyculata* | Ericaceae | Bilberry | Twigs |
| | | | | |
| **Herbaceous** | | | | |
| Graminoid | *Eriophorum vaginatum* | Cyperaceae | Cotton grass | Leaves |
| Graminoid | *Calmagrosti.* | Poaceae | Reed grass | Leaves |
| Graminoid | *Carex spp.* | Cyperaceae | Sedge | Leaves |
| Forb | *Cnidium  dubium* | Apiaceae | | Leaves |
| Forb | *Rubus spp.* | Rosaceae | Raspberry | Leaves |
| Fern | *Polypodium* | Polypodiaceae | Fern | Leaves |
| Fern | *Equisetum palustre* | Equisetaceae | Horsetail | Stem |
| Moss | *Sphagnum* | Sphagnaceae | Peat moss | Stem +leaves |
| Moss | *Polytrichum commune* | Polytrichiaceae | Hair moss | Stem +leaves |

**Table 2.** Summary of the mean aspect ratio, length (μm), surface area (μm$^2$), and mass retained (%) of charcoal produced in the muffle oven for individual plant species and fuel types from Siberia (SD=standard deviation).

| Scientific name | 250°C | 300°C | 350°C | 400°C |
|---|---|---|---|---|
| **Aspect ratio** | | | | |
| **Graminoids** | | | | |
| *Eriophorum vaginatum* | 7.2 ±5.0 | 11.1 ±6.4 | 5.8 ±3.7 | N/A |
| *Calmagrostis* | 7.4 ±4.6 | 12.9 ±11.8 | 6.0 ±3.8 | N/A |
| *Carex* | 7.4 ±5.1 | 10.6 ±7.0 | 8.3 ±6.0 | N/A |
| *Mean* | *7.3 ±0.1* | *11.5 ±1.2* | *6.7 ±1.4* | |
| | | | | |
| **Moss and ferns (whole plant)** | | | | |
| *Equisetum palustre* | 4.8±3.3 | 4.5±2.8 | 3.1 ±2.2 | 4.8 ±3.1 |
| *Polytrichum commune* | 4.8±3.1 | 4.2 ±2.6 | 4.2 ±2.6 | 4.7 ±2.9 |
| *Sphagnum* | 3.4±1.8 | 5.2±6.2 | 3.2 ±1.6 | N/A |
| *Mean* | *4.3 ±0.8* | *4.6 ±0.5* | *3.5 ±0.6* | *4.7 ±0.1* |
| | | | | |
| **Wood (trunk)** | | | | |
| *Pinus sylvestris* | 2.0±0.9 | 4.9 ±2.8 | 2.5 ±0.9 | N/A |
| | | | | |
| **Wood (tree twig)** | | | | |
| *Betula pendula* | 2.8±1.4 | 4.5 ±3.1 | 2.0 ±0.8 | N/A |
| *Picea abies* | 2.3±1.2 | 3.1 ±1.9 | 2.5 ±1.3 | N/A |
| *Pinus sibirica* | 2.4±1.3 | 3.1 ±1.9 | 2.6 ±1.3 | N/A |
| *Pinus sylvestris* | 3.2±1.7 | 3.5 ±2.0 | 2.9 ±1.7 | 2.9 ±1.5 |
| *Mean* | *2.5 ±0.4* | *3.8 ±0.8* | *2.5 ±0.3* | |
| | | | | |
| **Wood (shrub twig)** | | | | |
| *Chamaedaphne calyculata* | 3.5±1.9 | 6.3 ±3.5 | 3.7 ±2.1 | N/A |
| *Oxycoccus palustre* | 3.8±3.0 | 6.2 ±3.5 | 4.7 ±3.3 | 4.1±2,3 |
| *Ledum palustre* | 4.4±2.4 | 5.4 ±2.8 | 2.9 ±1.3 | 4.0 ±2.7 |
| *Vaccinium myrtillus* | 5.0±3.1 | 3.0 ±2.6 | 3.7 ±2.5 | N/A |
| *Mean* | *4.2 ±0.7* | *5.2 ±1.5* | *3.8 ±0.7* | *4.0 ±0.07* |
| | | | | |
| **Leaf (needles)** | | | | |
| *Picea abies* | 2.6±1.6 | 2.2±1.1 | 2.3 ±0.9 | N/A |
| *Pinus sibirica* | 4.0±2.3 | 3.6 ±1.8 | 4.7 ±2.9 | N/A |
| *Pinus sylvestris* | 4.0±2.5 | 3.7 ±4.0 | 3.5 ±2.9 | N/A |
| *Mean* | *3.5 ±0.8* | *3.1 ±0.8* | *3.5 ±0.1* | |
| | | | | |
| **Leaf (broadleaf tree)** | | | | |
| *Betula pendula* | 2.1±1.3 | 2.1 ±0.9 | 2.0 ±0.8 | N/A |
| | | | | |
| **Leaf (broadleaf shrub)** | | | | |
| *Chamaedaphne calyculata* | 2.4±1.3 | 2.3 ±1.7 | 1.8 ±0.7 | N/A |
| *Oxycoccus palustre* | 2.1±0.9 | 2.3 ±1.4 | 2.2 ±1.0 | 1.9 ±0.8 |
| *Ledum palustre* | 2.3±1.0 | 2.7 ±1.4 | 2.7 ±1.6 | N/A |
| *Vaccinium myrtillus* | 2.6±2.0 | 2.1 ±0.9 | 3.8 ±2.8 | 2.1 ±1.4 |
| *Mean* | *2.2 ±0.1* | *2.3 ±0.2* | *2.3 ±0.4* | *2.0 ±0.14* |
| | | | | |
| **Leaf (forb)** | | | | |

| | | | | |
|---|---|---|---|---|
| 765 | *Rubus* | 1.9±0.7 | 2.2 ±1.1 | 2.1 ±1.0 | N/A |
| | *Cnidium dubium* | 2.3±1.2 | 4.9 ±2.6 | 2.6 ±2.2 | N/A |
| | *Polypodium* | 2.1±1.0 | 2.9 ±2.0 | 2.5 ±1.3 | N/A |
| | ***Mean*** | ***2.1 ±0.2*** | ***3.3 ±1.4*** | ***2.4 ±0.3*** | |

**Length**


**Graminoids**

| | | | | |
|---|---|---|---|---|
| | *Eriophorum vaginatum* | 521 ±555 | 797 ±496 | 635 ±714 | N/A |
| | *Calmagrostis* | 620 ±622 | 951 ±712 | 440 ±370 | N/A |
| | *Carex* | 487 ±499 | 841 ±646 | 690 ±709 | N/A |
| 775 | ***Mean*** | ***543 ±64*** | ***862 ±79*** | ***588 ±131*** | |

**Moss and ferns (whole plant)**

| | | | | |
|---|---|---|---|---|
| | *Equisetum palustre* | 506±501 | 286±233 | 655 ±510 | 530 ±424 |
| | *Polytrichum commune* | 673±665 | 461±408 | 459 ±476 | 572 ±535 |
| 780 | *Sphagnum* | 612±574 | 524±636 | 319 ±217 | N/A |
| | ***Mean*** | ***598 ±84*** | ***423 ±123*** | ***477 ±168*** | ***551 ±29*** |

**Wood (trunk)**

| | | | | |
|---|---|---|---|---|
| | *Pinus sylvestris* | 408±347 | 391±300 | 482 ±384 | N/A |


**Wood (tree twig)**

| | | | | |
|---|---|---|---|---|
| | *Betula pendula* | 459±402 | 361 ±236 | 347 ±85 | N/A |
| | *Picea abies* | 435±371 | 439 ±303 | 598 ±575 | N/A |
| | *Pinus sibirica* | 450±379 | 318 ±248 | 427 ±343 | N/A |
| 790 | *Pinus sylvestris* | 593±558 | 379 ±318 | 407 ±295 | 407 ±350 |
| | ***Mean*** | ***469 ±72*** | ***377 ±44*** | ***452 ±95*** | |

**Wood (shrub twig)**

| | | | | |
|---|---|---|---|---|
| | *Chamaedaphne calyculata* | 387±340 | 347 ±288 | 525 ±528 | 521 ±555 |
| 795 | *Oxycoccus palustre* | 674±669 | 590 ±419 | 591 ±580 | 1053±700 |
| | *Ledum palustre* | 333±197 | 458 ±288 | 342 ±266 | 617 ±580 |
| | *Vaccinium myrtillus* | 461±446 | 818 ±714 | 441 ±306 | N/A |
| | ***Mean*** | ***463 ±150*** | ***553 ±202*** | ***474 ±107*** | ***730 ±283*** |

**Leaf (needles)**

| | | | | |
|---|---|---|---|---|
| | *Picea abies* | 549±535 | 342 ±276 | 608 ±692 | N/A |
| | *Pinus sibirica* | 606±587 | 385 ±292 | 432 ±414 | N/A |
| | *Pinus sylvestris* | 690±660 | 445 ±376 | 492 ±368 | N/A |
| | ***Mean*** | ***613 ±71*** | ***390 ±52*** | ***510 ±90*** | |


**Leaf (broadleaf tree)**

| | | | | |
|---|---|---|---|---|
| | *Betula pendula* | 493±498 | 335 ±234 | 354 ±185 | N/A |

**Leaf (broadleaf shrub)**

| | | | | |
|---|---|---|---|---|
| 810 | *Chamaedaphne calyculata* | 633±493 | 347 ±289 | 571 ±395 | N/A |
| | *Oxycoccus palustre* | 730±486 | 590 ±419 | 410 ±343 | 260 ±108 |
| | *Ledum palustre* | 728±698 | 393 ±322 | 568 ±586 | N/A |
| | *Vaccinium myrtillus* | 441±334 | 414 ±309 | 442 ±304 | 401 ±250 |
| | ***Mean*** | ***563 ±98*** | ***385 ±71*** | ***426 ±101*** | ***330 ±100*** |


**Leaf (forb)**

| | | | | |
|---|---|---|---|---|
| | *Rubus dubium* | 398±355 | 354 ±237 | 368 ±315 | N/A |
| | *Cnidium* | 466±384 | 372 ±263 | 549 ±598 | N/A |
| | *Polypodium* | 521±399 | 464 ±292 | 383 ±284 | N/A |
| 820 | ***Mean*** | ***466 ±61*** | ***418 ±65*** | ***466 ±117*** | |

***Surface area***

**Graminoids**

| | | | | |
|---|---|---|---|---|
| *Eriophorum vaginatum* | 37665 | 42744 | 51820 | N/A |
| *Calmagrostis* | 82058 | 99009 | 30244 | N/A |
| *Carex* | 32165 | 82947 | 51982 | N/A |
| ***Mean*** | ***50629*** | ***74900*** | ***44682*** | |


**Moss and ferns (whole plant)**

| | | | | |
|---|---|---|---|---|
| *Equisetum palustre* | 71381 | 25955 | 166741 | 75098 |
| *Polytrichum commune* | 123311 | 84632 | 68170 | 99522 |
| *Sphagnum* | 109185 | 77224 | 24724 | N/A |
| ***Mean*** | ***97346*** | ***62739*** | ***86545*** | ***87310*** |


**Wood (trunk)**

| | | | | |
|---|---|---|---|---|
| *Pinus sylvestris* | 105057 | 57028 | 97673 | N/A |


**Wood (tree twig)**

| | | | | |
|---|---|---|---|---|
| *Betula pendula* | 120539 | 37630 | 54180 | N/A |
| *Picea abies* | 92767 | 92107 | 199556 | N/A |
| *Pinus sibirica* | 119815 | 15058 | 83461 | N/A |
| *Pinus sylvestris* | 140004 | 74797 | 74929 | 88944 |
| ***Mean*** | ***115639*** | ***55051*** | ***102049*** | |


**Wood (shrub twig)**

| | | | | | |
|---|---|---|---|---|---|
| *Chamaedaphne calyculata* | 61732 | | 73382 | 131063 | N/A |
| *Oxycoccus palustre* | | 157873 | 51906 | 94418 | 249197 |
| *Ledum palustre* | 57191 | | 56836 | 254696 | 131860 |
| *Vaccinium myrtillus* | 72214 | | 291023 | 75212 | N/A |
| ***Mean*** | ***87252*** | | ***118061*** | ***136597*** | |



**Leaf (needles)**

| | | | | |
|---|---|---|---|---|
| *Picea abies* | 118375 | 86728 | 242364 | N/A |
| *Pinus sibirica* | 111068 | 46629 | 52880 | N/A |
| *Pinus sylvestris* | 134504 | 86639 | 83649 | N/A |
| ***Mean*** | ***121303*** | ***73332*** | ***126297*** | |


**Leaf (broadleaf tree)**

| | | | | |
|---|---|---|---|---|
| *Betula pendula* | 156591 | 67692 | 57934 | N/A |


**Leaf (broadleaf shrub)**

| | | | | |
|---|---|---|---|---|
| *Chamaedaphne calyculata* | 176620 | 73382 | 216473 | N/A |
| *Oxycoccus palustre* | 222081 | 184196 | 94939 | 31142 |
| *Ledum palustre* | 281751 | 67913 | 219788 | N/A |
| *Vaccinium myrtillus* | 88832 | 105547 | 88333 | 75360 |
| ***Mean*** | ***174636*** | ***87725*** | ***106408*** | ***53251*** |


**Leaf (forb)**

| | | | | |
|---|---|---|---|---|
| *Rubus* | 99109 | 79877 | 88927 | N/A |
| *Cnidium  dubium* | 108610 | 34207 | 176828 | N/A |
| *Polypodium* | 155561 | 106241 | 60473 | N/A |
| ***Mean*** | ***132085*** | ***70224*** | ***118650*** | |


**Mass retained**

| | | | | |
|---|---|---|---|---|
| *Eriophorum vaginatum* | 42.6 | 29 | 3.6 | 2.9 |
| *Calmagrostis* | 54.7 | 29.2 | 8.1 | 1.7 |


|  | | | | |
|---|---|---|---|---|
| *Carex* | 50.9 | 29.5 | 7.9 | 1.2 |
| *Mean* | *49.4* | *29.2* | *6.5* | *1.9* |

**Moss and fern (whole plant)**

| | | | | |
|---|---|---|---|---|
| *Equisetum palustre* | 54.4 | 49.4 | 31.0 | 21.4 |
| *Polytrichum commune* | 77.0 | 55.5 | 11.4 | 3.6 |
| *Sphagnum red* | 38.3 | 21.7 | 8.3 | 0.2 |
| *Mean* | *56.6* | *42.2* | *16.6* | *8.4* |

**Wood (trunk)**

| | | | | |
|---|---|---|---|---|
| *Pinus sylvestris* | 63.5 | 11.1 | 1.8 | 1.4 |

**Wood (tree twigs)**

| | | | | |
|---|---|---|---|---|
| *Betula pendula* | 50.8 | 40.2 | 10.9 | 0.3 |
| *Picea abies* | 56.5 | 40.1 | 16.8 | 3.4 |
| *Pinus sibirica* | 56.0 | 43.3 | 11.0 | 1.3 |
| *Pinus sylvestris* | 55.2 | 38.1 | 5.7 | 1.4 |
| *Mean* | *54.5* | *40.2* | *11.2* | *1.5* |

**Wood (shrub twigs)**

| | | | | |
|---|---|---|---|---|
| *Chamaedaphne calyculata* | 56.8 | 41.3 | 11.4 | 1.6 |
| *Oxycoccus palustre* | 78.8 | 49.4 | 35.9 | 14.8 |
| *Ledum palustre* | 56.6 | 52.5 | 15.6 | 0.9 |
| *Vaccinium myrtillus* | 60.8 | 42.2 | 16.9 | 2.5 |
| *Mean* | *62.7* | *46* | *20* | *4.8* |

**Leaf (needles)**

| | | | | |
|---|---|---|---|---|
| *Picea abies* | 67.2 | 52.3 | 26.2 | 4.9 |
| *Pinus sibirica* | 62.3 | 34.0 | 14.3 | 2.4 |
| *Pinus sylvestris* | 59.7 | 37.3 | 10.6 | 0.6 |
| *Mean* | *62.7* | *41* | *17* | *2.3* |

**Leaf (broadleaved tree)**

| | | | | |
|---|---|---|---|---|
| *Betula pendula* | 58.3 | 35.6 | 20.3 | 3.5 |

**Leaf (broadleaved shrub)**

| | | | | |
|---|---|---|---|---|
| *Chamaedaphne calyculata* | 61.8 | 44.7 | 27.2 | 4.3 |
| *Oxycoccus palustre* | 69.1 | 52.8 | 32.0 | 6.1 |
| *Ledum palustre* | 84.5 | 47.4 | 31.9 | 8.3 |
| *Vaccinium myrtillus* | 59.1 | 43.4 | 21.4 | 3.9 |
| *Mean* | *68.5* | *46.7* | *28* | *5.6* |

**Leaf (forbs)**

| | | | | |
|---|---|---|---|---|
| *Rubus* | 49.8 | 40 | 35.2 | 12.1 |
| *Cnidium   dubium* | 50.3 | 41.1 | 37.3 | 23.6 |
| *Polypodium* | 61.4 | 50.6 | 29.4 | 18.2 |
| *Mean* | *56.6* | *44.3* | *33.6* | *17.6* |

**Table 3.** The mean aspect ratio, length, surface area, and the number of each charcoal morphotypes in the Holocene samples ranging from 35 cm =185 cal yr BP to 303 cm=6750 cal yr BP from Ulukh-Chayakh mire. The local to regional vegetation is represented by the percentages of the main pollen types, whereas the local vegetation is represented by the plant macrofossils (values in brackets). Plant macrofossils are presented as numbers except for the wood remains presented as percentages.

| Depth (cm) | 35 | 84 | 85 | 172 | 248 | 268 | 303 |
|---|---|---|---|---|---|---|---|
| Aspect ratio | 3.2 | 3.0 | 3.0 | 4.0 | 11.1 | 4.3 | 2.8 |
| Length (μm) | 742 | 555 | 723 | 488 | 403 | 515 | 431 |
| Surface (μm$^2$) | 248.287 | 122.599 | 219.413 | 49.673 | 16.565 | 53.354 | 53.698 |
| **Charcoal morphologies (plant macrofossil)** | | | | | | | |
| Wood | 23 (6%) | 13(5%) | 6 (5%) | 11 (15%) | 1 (0%) | 8 (0%) | 6 (0%) |
| Leaf | 11 (0) | 7 (0) | 3 (0) | 2 (0) | 0 (0) | 4 (0) | 4 (0) |
| Needle | 0 (7) | 1 (0) | 0 (0) | 0 (0) | 0 (0) | 0 (0) | 0 (0) |
| Graminoid | 21 (30) | 9 (70) | 1 (70) | 11 (45) | 4 (65) | 19 (70) | 3 (0) |
| *Equisetum* | 15 (10) | 3(5) | 1 (0) | 0 (15) | 0 (10) | 5 (5) | 0 (90) |
| Moss | 10 (30) | 0 (5) | 0 (5) | 1 (5) | 1 (5) | 3 (5) | 1 (0) |
| **Pollen (%)** | | | | | | | |
| Trees | 90 | 92 | 92 | 94 | 97 | 92 | 97 |
| Shrub | 1.1 | 3.4 | 3.4 | 3.5 | 0.2 | 0.5 | 0 |
| Graminoid | 5 | 6.3 | 6.3 | 3 | 13 | 8.6 | 9.5 |
| *Equisetum* | 0.5 | 3.4 | 3.4 | 1.2 | 6.7 | 9 | 18 |
| Moss | 0 | 0.2 | 0.2 | 0 | 0 | 0.3 | 0 |

**Table 4.** Comparative results of the aspect ratio from plant species analysed in this study with those from literature. *Pinus sylvestris* wood sums the mean aspect ratio of wood from trunk and twig; wood total sums the mean aspect ratio of wood from trees and shrubs (trunk and twig); broadleaf sums the mean aspect ratio of leaf from trees and shrubs; whereas leaf total averages the mean aspect ratio of all leaf types.

| Fuel type | 250°C | 300°C | 350°C | 400°C | 500/550°C | open flame | References |
|---|---|---|---|---|---|---|---|
| Graminoid (boreal) | 7.3 | 11.5 | 6.7 | - | - | - | This study |
| Graminoid (artic) - | - | - | - | - | 6.7 | - | Pereboom et al. (2020) |
| Graminoid (forest steppe) | - | - | 3.6 | - | - | 4.8 | Umbanhowar & McGrath (1998) |
| Graminoid (grass) | - | - | - | - | 3.7 | - | Crawford & Belcher (2014) |
| *Pinus sylvestris* (wood) | 2.7 | 4.1 | 2.7 | 2.9 | - | - | This study |
| *Pinus sylvestris* (wood) | - | - | - | - | 2.8 | - | Crawford and Belcher (2014) |
| Wood (total) | 3.4 | 4.5 | 3.1 | 4.0 | - | - | This study |
| Wood (forest steppe) | - | - | 2.1 | - | | 2.3 | Umbanhowar & McGrath (1998 |
| Shrubs (wood and leaf) | - | - | - | - | 2.4 | | Pereboom et al. (2020) |

| | | | | | | | |
|---|---|---|---|---|---|---|---|
| Broadleaf | 2.2 | 2.2 | 2.3 | 2.0 | - | - | This study |
| Needles | 3.5 | 3.1 | 3.5 | - | - | - | This study |
| Leaf (total) | 2.5 | 2.7 | 2.6 | 2.0 | - | - | This study |
| Leaf (forest steppe) | - | - | 1.9 | - | - | 2.1 | Umbanhowar & McGrath (1998) |
| Leaf | - | - | - | - | 2.2 | | Crawford and Belcher (2014) |

**Appendix**

**Appendix A.** The watershed algorithm used to calculate morphometrics of charred particles.

The algorithm for automatic detection of morphometrics is based on functions from the Python module skimage (watershed algorithm). First, the picture is converted to a grey scale. A Sobel gradient of the picture is then calculated, which results in an elevation map. To use the watershed algorithm to detect the charcoal particles, a map of markers in the grey picture with grey values higher than 140 was then create. These are the starting points of the watershed region fill algorithm. Finally, any holes in the watershed regions were filled with the help of a binary fill method (Soille and Vincent, 1990). The detected particles were subject to the calculation of morphometrics such as surface area and lengths along the major and minor axis via supported functions. Particles with the length of major axis smaller than 150 µm were excluded from these calculations. The pixel area has been calibrated with a micrometre scale and the results scaled accordingly.

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

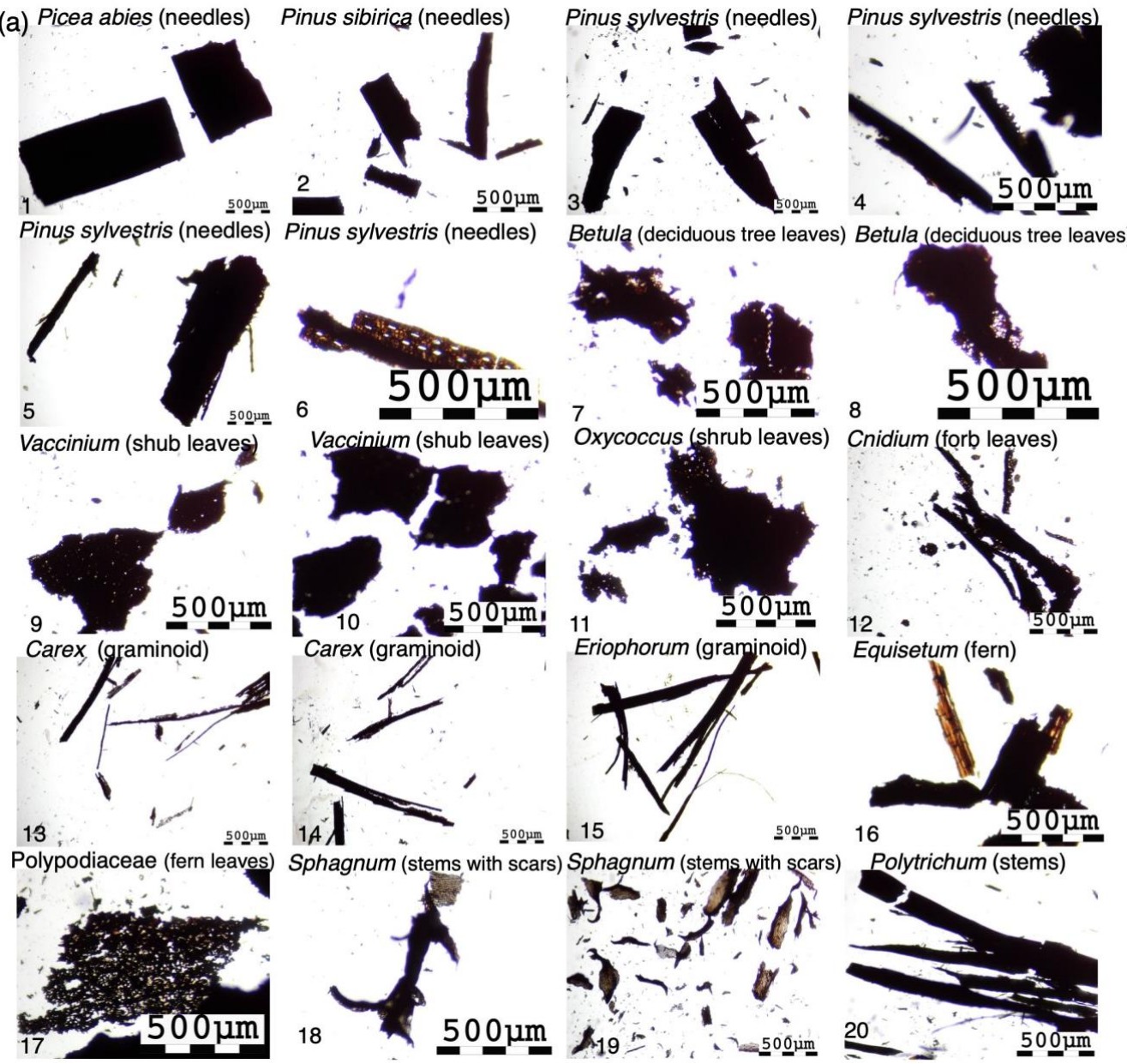

1015

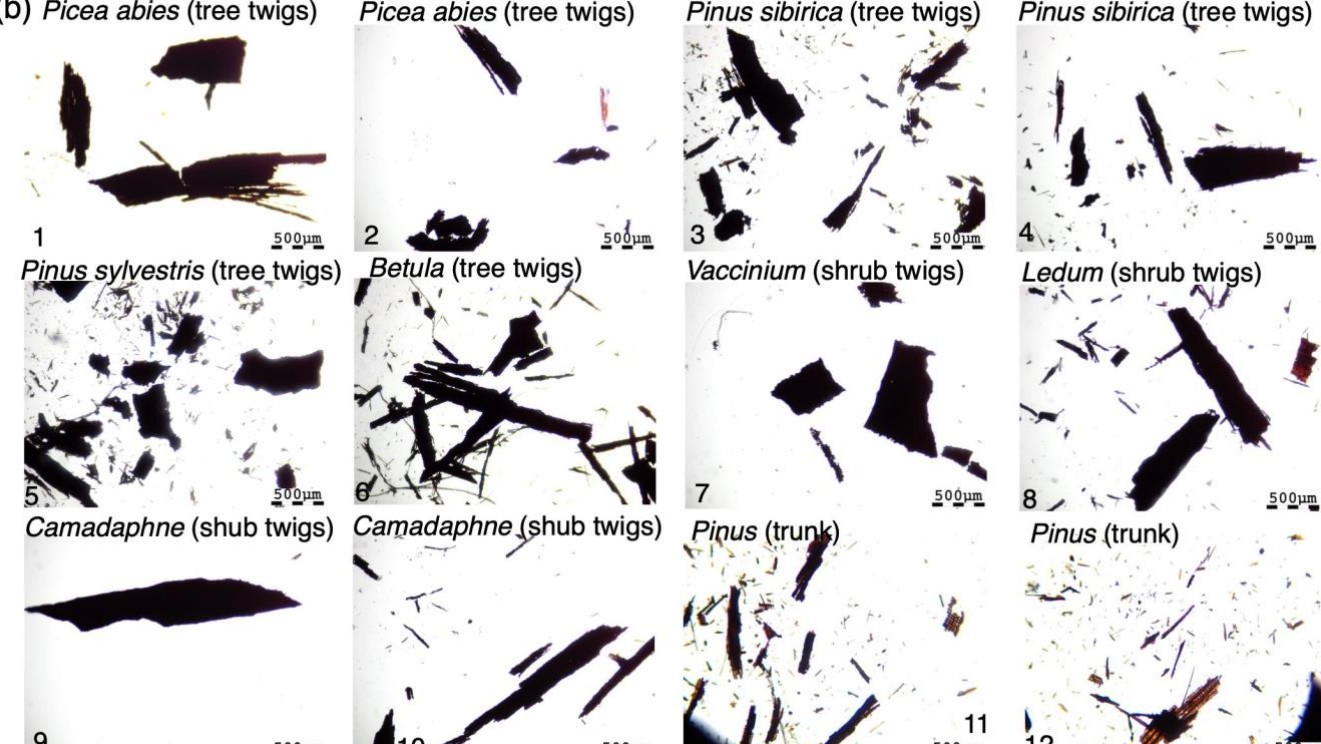

(b) *Picea abies* (tree twigs) — *Picea abies* (tree twigs) — *Pinus sibirica* (tree twigs) — *Pinus sibirica* (tree twigs) — *Pinus sylvestris* (tree twigs) — *Betula* (tree twigs) — *Vaccinium* (shrub twigs) — *Ledum* (shrub twigs) — *Camadaphne* (shub twigs) — *Camadaphne* (shub twigs) — *Pinus* (trunk) — *Pinus* (trunk)

1020