# Peer review of "Experimental production of charcoal morphologies to discriminate fuel source and fire type: an example from Siberian taiga"

_Biogeosciences, 2021_

## Author Response (AR1)

Frankfurt am Main, 7.05.2021

Dear Co-Editor-in-Chief and handling editor,

Please find enclosed a revised version of my manuscript entitled ''Experimental production of charcoal morphologies to discriminate fuel source and fire type: an example from Siberian taiga'', which contain a small alteration from the original title '' Experimental production of charcoal morphologies to discriminate fuel source and fire type in the Siberian taiga.

I have addressed and incorporated all comments made by the reviewers and the handling editor Sandy Harrison. In summary, I have:
(a) Provided more specific details about the methods and materials;
(b) Expended the discussion and placed this work in the context of previous studies;
(c) Expanded the measurements to fossil charcoal samples;
(d) Based on b) and c) expand the interpretation to palaeo-records;
(e) I have provided three new tables (Table 2, 3, 4) in the main paper, as well as in the SI (File S5) containing the morphometric measurements.
 I am fully aware of the need for such results for field practitioners, but I feel that it would be a stretch to say more on this based on current experiments. If necessary, I can delete this reference from the abstract.

I thank the reviewers for their valuable comments that helped to improve the current version of the paper and for encouraging words about this work, and I hope that they will approve the revised content of the manuscript.

*Reviewer 1 (see corrections in blue in the annotated manuscript) This is an interesting and useful combination of field work observation, sampling and lab experimentation to examine modern plant communities and charcoalized plant parts at different roasting temperatures for the purpose of improving the development and interpretation of palaeoenvironmental records of charcoal (a proxy for fire) in boreal forests. The results should be of interest to a wide variety of researchers in EGU fields and interested in biogeosciences related to fire at local to large spatial scales, palaeovegetation research, and archaeology.*
*OVERVIEW: Overall I suggest some changes to clarify the intent of the author for certain words like "burned", 'impact', 'intensity' etc - detailed below.. And some expansion on the discussion with respect to additional charcoal morphology literature in paleoenvironmental studies and archaeological studies. There should be some refinement on how explicit the 'decomposition' technique of the vegetation fuels was and thus, the use of the word ''burned''. Is it known if the material flamed in the oven? Was it roasted? combustion? Pyrolysis? The oxygen and time variables are largely ignored and this should be written as a caveat in the experimental design to be explored elsewhere or in future studies. Similarly, the words charcoal, charred, and (roasted, unused term in the study) may need to be defined early. 'Ashed' is also used but not fully discussed - I suppose it meant a more complete combustion generating white ash that then crumbled apart into soot and flyash? Was there an explicit purpose statement? Something like, characterising the diversity of charcoal morphologies produced by boreal forest vegetation fuels at X study site?*
R: Many thanks for the positive response that this paper was an interesting and useful contribution to the community, and for the useful and thoughtful comments to further improve this paper. In revising this manuscript, I have 1) Clarified the terms used and expanded on the caveats of experimental design; 2) Expand the discussion concerning additional charcoal morphology literature in paleoenvironmental studies including comparisons with those focusing on transport and sorting of morphologies. However, a comprehensive comparison with archaeological studies is beyond the scope of this study. For a detailed list of adjustments please my response below.

SPECIFIC COMMENTS:
*L36 - This might be semantic, or a question of (spatial and time) scale and thus the need for clarity. The word impact is a bit ambiguous without further clarity on the context and use of this term. What is meant here '[a] fire impacts boreal forest''. Over the long duree, is it the changing attributes of fires and the fire regime that impact the boreal forests? Does this mean one fire is impactful? Boreal forests have a lot of spatial heterogeneity in vegetation structure that is in part caused by fire and in part also influences fire itself. A changing fire regime has significant outcomes on the land cover. But if fire is a process in boreal forests itself, it seems more of a feature than of the biome rather than something that just impacts it. Throughout the paper the framing of the disturbance regimes needs to be balanced with how these disturbance regimes (mostly fire explored here) are a part of the system, and not something that just happens to the boreal forest and changes (impacts) it.*
R: Thank you. I have rephrased this to acknowledge the role of disturbance by fire in the functioning of boreal forests, but also the concerns on the impact of changes in the fire regime. It reads: ''Disturbance by wildfires is

among the most common disturbance types in boreal forests, triggering gap dynamics or stand-scale forest replacement depending on intensity and frequency (Goldammer, 2015). Ongoing and anticipating increase in the intensity and frequency of wildfire in boreal forests is raising concerns on its impact on the composition of these forests as well as climate (Jones et al., 2020).''. L 34-37.

*L40 - Fire intensities in nature have been shown to be able to reach much higher temperatures, even flame temperatures can be higher than the range explored here. Is this really the gamut of temperatures in hot boreal fires? This needs to be framed as a subset (or modal?) temperatures of fires (maybe this can be estimated from MODIS intensities? i.e. energy output detected by satellite, or if there are some published field-based measurements.).*
R: Wildfire burns at higher temperatures than 500 degrees, but with this sentence, I highlight temperature ranges that lead to the charcoal formation. I have revised this sentence to reflect the two points: 'Wildfires reach temperatures up to 1800 °C, however, charcoal is an inorganic carbon compound resulting from the incomplete combustion of plant tissues, which typically occurs at temperatures of 280–500 °C (Rein, 2014).'' L. 43-44.

*L49 - worth stating somewhere that the Courtney-Mustaphi and Pisaric, 2014 study discussed potential for not just focusing on known-fuel morphotypes for charcoal analysis but for categorising all morphologies found in a local-scale study to examine the variability; as this would be useful to explore relationships to not just the known-fuel-sources of charcoal but taphonomic processes and possibly fire types (or another variable).*
R: Revised: ''Courtney Mustaphi and Pisaric (2014a) also discussed the potential for categorising of charcoal morphologies to explore relationships to taphonomy processes and fuel types''. L. 70-72.

*L60 - It would be useful to make distinctions between studies using ovens, flames, and other pyrolysis and combustion methods.*
R: Revised. Please see l. 60-66, chapter 4.2, and Table 4.

*L68 - spp? Or taxa? What was the minimal taxonomic resolution?*
R: Genus is the minimum taxonomic resolution used here, however, most plants have been identified at species level. Please see Table 1.

*L69 - was there any testing in this study? It appears to be mostly a characterization study, which has merit. The purpose, objective, aims are not congruent with the content*
R: Revised ''This paper presents the first results of laboratory-produced charcoal morphologies (muffle oven) spanning a range of fuel types originating from 17 boreal Siberian species. It aims to characterise the diversity of charcoal morphologies produced by boreal understory and forest vegetation to facilitate more robust interpretations of fuel sources in the study region. Specifically, it evaluates (i) whether morphological distinctions (morphometrics and finer anatomical features) exist between species and fuel types, (ii) the effect of burning temperature on the mass, morphometrics, and finer anatomical features of charred plant material, and (iii) discusses the advantages and limitations of laboratory-based burning studies for palaeofire reconstruction.'' Please see l. 75-80.

*L79 - complete dryness. Was this checked? Before burning in the oven were the samples dried? Often one would dry at 105°C for 24 hours to drive off most moisture. Of course this may only influence the combustion to a limited extent in this study - but worth documenting for future comparison studies.*
R: Thank you. I removed the word completely dryness. The material was air-dried for several weeks, but the moisture content was not measured. The plant appearance was dry and brittle in all instances.

*L87 - what was the rationale for limiting oxygen? Were there any comparisons with oxygen not-limited burning and open flame burning?*
R: To slow the burning process and better replicate the natural conditions where fuel is more abundant and layered, and thus the oxygen is more limited. I have not attempted experiments under non-limiting oxygen nor open flame conditions, but Umbanhowar and McGrath, 1998, found small differences between the muffle oven and the open flame burning.

*L198 - is Ericaceae ever investigated in this study?*
R: *Ledum, Vaccinium, Camadaphne, Oxycoccus* are all part of Ericaceae family, this is what I mean here, however, I have added now their full names (see also Table 1.

*L231 - ''rounder'', was this intended to mean circular? (as in 2-Dimensional), or roundedness as in the degree angles are eroded or not produced? Can these terms be written more explicitly for the reader. Note that both how*

*circular something is, and roundedness can be quantified, semi-quantified or categorised. Was this done? Discussing if this may or may not be useful in future studies would be useful for readers and future analysts (Note Vanniere et al 2003 Journal of Archaeological Science, 30(10), pp.1283-1299, with reference to eroded charcoal in agricultural soils).*

R: Rounder is intended as circular, as in 2-Dimensional scale. I have not quantified the degrees of angels. In the revised version I have introduced a chapter that discusses the influence of particles shape on transportation by air and water including the potential usefulness of additional measurements such as roundedness. Please see 4.2 l. 298-322.

*L264 - Add a caveat about the need to do detailed comparative studies on graminoid versus conifer needle fuels and subsequent charcoal. And perhaps among Graminoid growth forms themselves: Poaceae subfamilies, Cyperaceae, and others.*

R: Added. 'Comparative studies on graminoid charcoal originating from Poaceae (grass) versus Cyperaceae (sedge) family will further improve the identification of fuel types given the ecological differences of the two groups i.e., sedges growing on wetlands, and grass often on dry habitats. L. 229-330.

*L279 - a useful document for comparing mosses etc for readers to compare in Quaternary and temperature ecosystems is Levesque et al 1988. Lévesque, P. E. M., Dinel, H., Larouche, A. 1988. Guide to the identification of plant macrofossils in Canadian peatlands. Land Resource Research Centre, Research Branch, Agriculture Canada, Ottawa.*

R: I have cited Grosse-Brauckman, 1972 and Schweingruber, 1978.

*L315 - intensity, as in heat/energy given off by fire?*

R: Adjusted to denote intensity as high-energy, whenever I refer to temperature of fire

*General comments: Introduction in general:I think there needs to be a distinction between flame combustion, roasting by hot air (ovens), pyrolysis. This needs to come out more obvious to the read beginning in the abstract, methods, and discussions. It needs to be stated that dry roasting in an oven is a proxy for one type of heating of vegetation in a natural fire, different to flame burning, etc. This is evident in the statement by the authors on L124 that 'All plant tissue was reduced to ash at 450 °C (Fig. S1).' In natural fires, flame and air temperatures do reach higher. I think the main items that need to be acknowledged is that the oven approximates some aspects of the heating conditions of natural fuels and that a crucial variable that is not explored is time at a (burning) temperature. With roasting in an oven the influence of flame dynamics and turbulent air flow is missing to the same degree as fires outdoors. This needs to be acknowledged as part of the experimental design and open the need for additional research.*

R: Done. As this comment repeats the one at the beginning of please see my reply there.

*It would be useful throughout and within this paper (if anything was combusted in a different method) to add the categorical naming of how the material was 'burned'. See Table1 in the following publication: https://doi.org/10.1016/S0031-0182(00)00174-7*

R: All plant material used for the current burning experiments was dried before and burning conditions (muffle oven, preheating) were the same for all measurements. The only difference is the use of different temperatures. This information was added in the caption of **Table 1**. List of plant materials burned.

*I have some broad suggestions on how certain details are communicated.-The plant anatomy of bryophytes is treated rather colloquially and requires refinement. - Are the species names known for the bryophytes? Many burn differently at low temperature because they hold water droplets differently, making some taxa more difficult to ignite even under the same fire weather conditions.*

R: I have used *Sphagnum* (likely S. medium/S. divinum) and *Polytrichum commune* (brown moss). Their names are visible in all graphs and Table 1. The comment that moss species hold water droplets differently, making some taxa more difficult to ignite is pertinent and holds in nature, but not in laboratory burning experiments where all plants were dried prior to combustion. Extending this interpretation would be a stretch.

*-The use of the word 'twig' needs some level of description here as twigs are different in deciduous, coniferous, herbaceous? and colloquial terms. Can this be more explicit throughout the paper as it may vary by plant types.*

R: The term twig was used only for woody species i.e, deciduous and coniferous trees as well as deciduous shrubs, and denotes small branches close to where the leaves are attached to the plant. The caption of Table 1 was extended to reflect the plant types used.

*-There is a lot of comparison with Mustaphi and Pisaric 2014; could this be expanded to many of the other morphology papers. A table on charcoal morphometric technique studies and the usefulness could help link with the editors comment on this study not presenting a tangible application of the study in its current form.*

R: To accommodate this and the comment from Rev 2, I have greatly expanded the discussion on morphometric results on chapter 4.2 and added extra tables summarizing morphometric results (aspect ratio, length, surface area) from this study (Table 2), the full range of measurements (File S5) and comparative aspect ratios (this was the most employed morphometric across studies) from publications (Table 4). I have also discussed implications of charcoal morphometric for the reconstruction of fuel-type and transportation by air and water.

FIGURES:

*-Perhaps the black font text would be best placed outside the photograph because of the overlap and poor contrast between the letters and charcoal fragments.*

R: Thank you, I revised the figures accordingly.

*FIGURE4 - are some of these not charcoal? Again the Levesque et al 1988 publication might be worth comparing.*

R: All pictures come from laboratory (muffle oven) produced charcoal.

*FIGURE5 - 'chacoal production' spelling in bold (bottom left). Can you quantify the aspect ratios?*

R: Corrected to charcoal production. Aspect ratio=L:W ratio.

*TABLE1 - can you add growth forms of plants? (sort of in the plant type column) and the anatomical parts investigated in this study? For instance, Does ''leaves'' include the Petiole? The veins? Does twig also just mean wood? Or something else? Soft young wood? High water content?*

R: The growth form of plants is now included in the figure caption.

*ADDITIONAL PUBLICATIONS*

*Some important morphology studies are not discussed in the context of this study. It would be appropriate to discuss these studies in a comparative manner and to build the case for the overall usefulness of morphological metrics.*

R: Many thanks for providing such an extended listed of papers, the vast majority is now incorporated into the revised paper.

**Reviewer 2** (*see corrections in violet in the annotated manuscript)*
**General Comments:***: https://doi.org/10.5194/bg-2021-1-RC2*

*Feurdean presents a dataset of experimental charcoal produced from 17 species endemic to boreal Siberian. Using these experimentally produced charcoal, Feurdean makes insights into the reliability and applicability of charcoal morphologies as a proxy of fuel type. Additionally, the author shows how charcoal mass is retained as a function of combustion temperature for these samples.*

*The manuscript is very interesting and presents a promising dataset for the paleofire field. Its efforts towards proxy calibration of charcoal morphology and morphometry represent a key advance. Its experimental characterization of charcoal produced from several new fuel types and taxa, as well as replication of previous experimental productions of charcoal, make it a useful contribution to the field. Mass retention during charcoal production represents a key gap in our understanding of the source to sink controls of sedimentary charcoal, and this manuscript helps to bridge this gap. Lastly, it improves on some of the methodological approaches of earlier experimental productions of charcoal particles.*

R: I want to thank the reviewer for the useful and thoughtful comments, as well as the overall positive response in feeling that this paper was an interesting and useful contribution to the community.

*However, the manuscript falls short in several ways. Although the bulk of the manuscript is in good order, and the study itself is robust and scientifically sound, the discussion and conclusions are, in my opinion, woefully underdeveloped, especially in light of the novelty of the dataset and approach. In a broader sense, the manuscript fails to fully deliver on the potential conclusions and insights that could be gained from a dataset which is truly*

*brimming with potential. I recommend moderate revision prior to publication and have outlined, in my opinion, the manuscript's primary shortcomings and areas needing improvement below.*

*Firstly, it lacks in-depth comparison to prior work, which will surely undermine its impact. For example, several of the taxa tested by the author have also been directly tested in previous experimental studies (e.g., Eriophorum vaginatum in Pereboom et al. (2020), Pinus sylvestris in Crawford and Belcher (2014)), yet there is no discussion of the similarities and differences of the morphometrics of the charcoal produced from these taxa. Similarly, previous experimental studies have used a variety of techniques (and temperatures) to produce charcoal, but limited comparison is made with these studies and their conclusions. How and why do the values differ and compare between this and other experimental charcoal studies? Although the Discussion focuses on the findings of this study, it does not sufficiently contextualize these findings within those of the published literature. For example, section 4.2 refers to several other studies, but only vaguely compares findings of charcoal particle elongation between these studies. The published aspect ratio data of these experimental studies should be more thoroughly discussed and explored if this is this to have a veritable impact on the field.*
R: To accommodate this and the comment from Rev 1, I have greatly expanded the discussion on morphometric results on chapter 4.2 and added extra tables summarizing morphometric results (aspect ratio, length, surface area) from this study (Table 2), the full range of measurements (File S5) and comparative aspect ratios (this was the most employed morphometric across studies) from publications (Table 4). I have also discussed implications of charcoal morphometric for the reconstruction of fuel-type and transportation by air and water.

*Secondly, the manuscript does not make actionable conclusions for the paleofire field. Besides a somewhat unconvincing (see specific comments below) description of the potential utility to use charcoal aspect ratios to distinguish fire and fuel types, the manuscript does not provide explicit descriptions of the morphometric values that be used to constrain interpretations of sedimentary charcoal. What is the cut-off for elongation indicative of graminoids? What are the ranges of morphometric values that can be indicative of fuel types? What are the mean values of the aspect ratios of the fuel types that can be distinguished (wood, graminoids, and leaves, as indicated in the Conclusions section)? What is the quantitative relationship between temperature and charcoal mass retention? More specific and worthwhile conclusions need to be made from the dataset. At present, the manuscript is intriguing but does not provide explicit values and tools that can be applied to actual sediment samples, and in turn, inform paleofire interpretations.*
R: As stated above I have extended the Discussion to a) the range and mean morphometric values and, where possible, the cut-off values indicative of fuel types and influence of particle shape on transportation by air and water (4.2); b) the quantitative relationship between temperature and charcoal mass retention (4.1); c) extended the work to the morphometrics and morphologies of fossil charcoal particles (4.4); and d) Applications and future recommendations (4.5). However, the overlapping aspect ratios make it difficult to defined cut-off values for many fuel types (see also Crawford and Belcher, 2015).

*Lastly, the author should provide the morphometric values derived from the experimental charcoal in Table 1 (or in the supplement) to enable others to more directly compare with this dataset. As it stands, future work will have to estimate values from the figures. To my knowledge, provision of explicit data value ranges is the norm in these types of studies. I strongly suggest the author provide these values to facilitate use of the insights provided by the manuscript.*
R: Please see the new Table 2 with the mean values of aspect ratio, Length, and Area in the main text and the new Table (File S5) with the full range of individual morphometric measurements across all temperatures in the Supplementary Material.

*Specific Comments:*
*L103-104: What was the rationale for this sieve size, given the wide range of sieve sizes used in the paleofire field?*
R. To get rid of very small particles, but this step could be skipped.

*L203-206: Shouldn't this be irrelevant given that the sampled were uniformly dried before combustion?*
R: I have removed this sentence from here and added it at the methods to point out that while all other material collected was from alive plants, the trunk wood comes from a dead tree. L. 93-94.

*L243: Neither of these citations are provided in the references list. The Clark (Clark 1988) and Higuera (Peters and Higuera 2007, Higuera et al. 2007) models of charcoal dispersal do not actually incorporate particle shape. To my knowledge, the model of Vachula and Richter (2018) is the only one to directly test the effect of charcoal particle shape on dispersal distance.*

R: I have added these references into the reference list and extended the chapter 4.2 to reflect the the influence of particles shape on transportation by air and water (4.2). Please see lines 296-319.

*L248: Aleman et al. did not experimentally produce charcoal particles. The values referred to here were derived from environmental samples. This comparison is not appropriate, in my opinion.*
R: I have retained this reference but make it clear it comes from fossil records.

*L297-311: This conflation of fuel type and burn temperature is not convincing. Although the data presented in this paper clearly show the ability to differentiate fuel types, it seems a stretch to suggest that burn temperature might be inferred from fuel type assemblages and charred mass. How would this work for an environmental sample? How could charred mass be differentiated from total fire activity? Why isn't Figure 2H referred to and discussed in this section? It should be useful in making these conclusions.*
R:The differentiation between fuel types is not straightforward and cannot be made based on charred mass alone, but by additionally looking at charcoal morphologies of various fuel types. For example, if we know that Sphagnum preserves as charcoal at lower temperatures and if we find abundant Sphagnum charcoal morphotypes in the sediments, this could potentially mean low temperatures of fires, otherwise Sphagnum would not have been preserved as charcoal. In the revised version of the manuscript, I have extended the morphometric and morphological measurements to the fossil charcoal at the nearby peatland (Teguldet village), which in combination with results from experimentally produced charcoal are used to infer fuel type and to some extend fire types. Please see the revised Chapters 4.4. and 4.5. Fig2H shows the aspect ratio for fuel mix at a single temperature and has been used to determine the accuracy of aspect ratio in mix fuel types, as normally the case in nature.

*L359: Where are the morphometric measurement data? How can future researchers actually use these data to better inform their interpretations if they are not provided?*
R: All the morphometric measurement data are now provided in the paper (Table 2) and Supplementary Material. (SI4).

*Figures 2 and 3: The figure captions indicate that the boxplots summarize the median aspect ratios, lengths, and areas of particles for each taxa (i.e., each box plot depicts the median, standard deviation, and range of the median values of the measurements). If I understand correctly, though, these boxplots are actually summarizing the individual measurements. The medians are just one component of the boxplots? This ought to be clarified.*
R: Done: ''The median aspect ratios of charred particles from (a–d) the individual measurements taxa burned at 250, 300, 350, and 400 °C, respectively, and (e–g) fuel types at burning temperatures''.

*Technical and typographical corrections:*

*L14-15: "Graminoids, Sphagnum, and wood"*
R: Corrected.
*L38: End parenthesis is missing from citation.*
R: Parenthesis included.
*L48: Consider rephrasing. No fires are 'cool'. Consider using more specific fire regime characteristics (e.g., intensity, severity).*
R: We have replaced cool fires with low intensity or low temperature fire throughout the manuscript
*L71: "paper" singular*
R: Done.
*L85: "tests"*
T: Done.
*L248: "Mustaphi"*
R: Mustaphi and Pisaric (2014)
*L287-288: "influence"*
Done
*L292: "cool, surface fueled"*
R: Done
*L293: "Anderson"*
*R: Done,* Anderson, 1982
*L297-300: Please consider revising this sentence. Its present wording is difficult to decipher.*
R: Note applicable any more as this chapter was restructured.
*L310: "reveal"*

R: Done
 R: All reference were included in the reference list

*RC3 ( see corrections in red in the annotated manuscript)*

*'__Comment on bg-2021-1'__, Anonymous Referee #3, 16 Mar 2021*

*General comments*
*This manuscript provides new and additional information to the increasing body of knowledge of using sedimentary charcoal for more robust reconstructions of past fire regime including fire severity, fuel sources and fire type. The study combines different features analysed from modern burned plant materials and based on that provide assessment of possible methodology for defining fire type and fuel source also from sedimentary records. It would be interesting to see a comparison of the features from plant material burned in laboratory with features from actual sedimentary charcoal record from the same study area. However, I understand that this could be further work in addition to this manuscript. In general, this manuscript is well organized and written in clear language. Introduction is informative and the aims of the study are clearly stated. Methods used in the study are relevant and justified and the approach provides novel information and methodology for using sedimentary charcoal in reconstructing past fire regime in boreal landscape. The results are interesting and I would have hoped to see more comparison of the results from this study to previous research also using laboratory techniques or sedimentary charcoal to analyze charcoal morphology in regards to the combustion temperature and fuel source. I was also partly missing clear numerical values when describing the morphological features. For example, I would have hoped more defined information of what is the size range of smaller and larger fragments, when morphometrically categorizing fuel types. More exact numerical information of size, ratios, mass retention etc. would make these results more comparable with later studies utilizing the methodology presented here. Overall, I think this work brings important addition to the literature and methodology of using combination of different charcoal features for more robust reconstructions of past fire regime. I have* made some minor suggestions for the author in the detailed comments. *In my opinion, this paper would be in wide interest of the readers of Biogeosciences and I recommend this paper to be accepted to publication after suggested minor revisions.*
R: Many thanks for the positive response that this paper was an interesting and useful contribution to the community, and for the useful and thoughtful comments to further improve this paper. Following your suggestions and that of reviewers 1 and 2, I have a) expanded the discussion on how the results from this study compared to previous research using laboratory techniques to determine the combustion temperatures and fuel sources; 2) included tables containing numerical information on aspect ratio, length, and area of charcoal particles; and 3) extended the work to the morphometrics and morphologies of fossil charcoal particles.

*Detailed comments:L 79: I would suggest adding right in the beginning that these identified plant materials were the actual samples that were later burned in the laboratory.*
R: Added: ''Plant materials used for laboratory burning experiments were identified in the field, stored in plastic bags for transportation, and air-dried''. L. 88-90

*L 87-88: So, nothing was actually used to initiated flame, but the burning was due to high temperatures? How well does this mimic the natural conditions for fire and does it have an effect on the charcoal features compared to the ones in sedimentary record?*
R: Right, nothing was done to initiate the flame. Laboratory conditions are not identically to the natural wildfires and I have acknowledged this (see l. 106-110). Results from charcoal morphometrics obtained under open flame compare well to those in muffle oven (see Umbanhowar and McGrath, (1998) and charcoal features obtain in the laboratory with those from sediment wildfire (see Mustaphi and Pisaric (2014a).

*L 92-95: I would suggest to mark the mixed samples a bit more clearly. Now it takes some time to figure out that which proportion go to which sample.*
R: The samples are presented along a gradient of transition from predominant surface fuel fire that burns with lower intensity (graminoid and moss), to surface fuel that burns with intermediate- to high-intensity (with additional shrubs), and finally to crown fuel that burns at high-intensity (wood and tree leaf). I agree that the marking is complex but so is the combination of fuel types in nature. I have clarified this in the figure caption but retained the marking in the figure itself.

*L 126-128: Here is reference to the Fig. 1. However, there isn't as many types given in the figure as here in the text. This is a bit confusing and I recommend to fix this or adding some explanation for leaving some features out form the figure.*

R: Many thanks. The description of Fig. 1 in the text follows the groping of similar fuel types at the bottom of this figure. In the revised manuscript I have better matched the names of these groups in the Fig. 1 with those in the text: ''The average percentage of charred mass retained at 300 °C (an intermediate temperature) was as follows in decreasing order: brown moss and fern (50%) > wood (shrub twig 46%) > leaf (shrub 44%) > leaf (forb 42%) >leaf (needles 41%) > wood (tree twig 40%) > graminoid (29%) > *Sphagnum* (22%) > wood (trunk 11%; Table 2; Fig. 1)'' Please see lines 143-144.

*L 197-199: Here average results across all temperatures are referred to Fig. 1, but as far as I understand the figure presents results from different burning temperatures rather than averaged over all temperatures. I would suggest this to be clarified.*

R: I have revised the text to better illustrate Fig 1. ''Graminoid, *Sphagnum*, and trunk wood produce the lowest amounts of charcoal per unit biomass and lost their mass more rapidly with increasing burning temperature i.e., from 40-63% at 250°C to 0.2-3% at 400°C (Fig. 1; Table 2). Contrastingly, leaves of heathland shrubs, forbs, and ferns (Polypodiaceae), as well as fern stems (*Equisetum*), produced the most charcoal per unit biomass and retained the greatest mass at higher temperatures i.e., from 50-84% at 250°C to 4-24% at 400°C (Fig. 1).'' Lines 217-221.

*L 235-237: Here it is stated that larger fragments are more reliable to categorise fuel types. It would be useful to clearly state that what is the size ranges for what is considered larger and smaller fragments.*

R: Indeed, in the original manuscript I have stated ''Because smaller charred fractions tend to be rounder (lower aspect ratio) than larger fractions, it suggests that larger charcoal fragments can be more confidently used to morphometrically categorise fuel types''. In fact, results from this study show that the smaller the particles, the smaller the aspect ratio was. From here I have extrapolated that larger charcoal fragments can be more confidently used to morphometrically categorise fuel types. However, it is difficult to find a clear cut of the actual size that could more reliably be used for aspect ratio, therefore this sentence now reads 'Although larger charcoal fragments may be more suitable to categorise fuel type, it is difficult to define a threshold aspect ratio concerning the size of the particles to be used for such measurements' Lines 257-259.

---

## Author Response (AR2)

**Co-Editor-in-Chief Decision: Publish subject to minor revisions (review by editor)** (25 May 2021) by Kirsten Thonicke

*Dear Angelica,*
*I agree with the minor revisions suggested by Sandy.*
*Best wishes,*
*Kirsten*

*Associate Editor Decision: Publish subject to minor revisions (review by editor)* *(21 May 2021) by Sandy Harrison*
*Comments to the Author (doc): bg-2021-1-comments-to-author.doc*
*Comments to the Author:*

*Dear Angelica,*

*Thank you for your response and for addressing the reviewers' comments so thoroughly in your updated manuscript. The addition of the new tables with information about the experimental results is very welcome. The expanded discussion of previous work is also useful. The revised and expanded manuscript is much improved, and I am happy to accept this subject to minor editorial revisions. Please could you reorganise Table 2 to make it more immediately readable. I think you could do this by organising it so that all of the measurements made on a single species are in a single row. Also by having the PFT as a separate column rather than a heading. (See attached suggested layout in the attached word file).*
*While it is important to keep Appendix A in the main text, Appendix B should be moved to Supplementary online material. Figure 4 should be moved into this Supplementary. The clarity of the text could be further improved. I have made some suggestions for correcting and simplifying the text in the attached file (in red), but are not compulsory. I hope that these suggestions will help you revise the manuscript. Sandy*
*P.S. You said in your response that you didn't quantify the degree of angels, which is hardly surprising given there are so few of them about nowadays.*

Dear editors,

Please find enclosed a revised version (2) of my manuscript entitled ''Experimental production of charcoal morphologies to discriminate fuel source and fire type: an example from Siberian taiga''.

I have incorporated all the additional minor comments made by the Associate Editor, which includes:
(a) Made all the linguistic corrections she has suggested. I very much appreciate her effort to make these improvements.
(b) I have re-organised the layout of Table 2 as she suggested i.e, made all the measurements of a single species and fuel types in a single row.
(c) I have moved all the pictures of macrocharcoal (Fig.4a,b,c) and microcharcoal (Appendix B) morphologies into the Supplementary online material. Fig.4a,b,c is now File S5a,b,c, whereas Appendix B is now File S6a,b.

Many thanks for these additional suggestions to improve the manuscript, and looking forward to seeing this manuscript published.

Kind regards
Angelica
Department of Physical Geography, Goethe-University
Altenhöferallee 1,60438 Frankfurt am Main
angelica.feurdean@gmail.com
Feurdean@em.uni-frankfurt.de
Telefon 49 (0)69 798 40166